# Bubble explosion induced melt pool instabilities in electron beam melting of aluminum alloy

Jiandong Yuan [1,2], Luis I. Escano[1,2], Samuel J. Clark [3], Junye Huang [1,2], Ali Nabaa[1,2], Qingyuan Li [1,2], Kamel Fezzaa[3] & Lianyi Chen [1,2] ✉

Electron beam melting (EBM) is an additive manufacturing technology that can process materials and manufacture components otherwise impossible or uneconomical. However, defects, including porosity and surface irregularities, are widely reported in EBM-built components, and their formation mechanisms are not fully understood. Here, using in-situ high-speed synchrotron X-ray imaging, we reveal that bubble explosions in Al6061 during EBM induce melt pool instabilities contributing to defect formation. The melt pool and keyhole evolve through three stages: (1) initial formation of a melt pool, (2) subsurface bubble formation and explosion, and (3) periodic keyhole oscillation. During scanning, periodic bubble explosions can eject molten liquid as spatters and disturb the vapor depression and melt pool, contributing to surface humping, that may trigger lack-of-fusion defects in subsequent layers. The physical insights we report could provide guidance for EBM machine development, process innovation, alloy design and model development.

Electron beam melting (EBM) additive manufacturing process utilizes a focused electron beam to selectively melt and fuse feedstock powders layer by layer to create high-value parts within a vacuum chamber. It has emerged as an important technology in advanced manufacturing, offering advantages of high energy efficiency for processing highly reflective metals (e.g., pure copper for heat exchangers and electrical components)[1–4], low residual stress for processing brittle materials (e.g., TiAl for jet engine blades)[5–10], clean environment for processing impurity-sensitive materials (e.g., tungsten and refractory high entropy alloys for fusion energy applications)[11–15], and fast beam steering for programmable thermal control[16,17]. However, the adoption of EBM for mission-critical components remains constrained by processing defects, including porosity, surface irregularities and spattering, which compromise part quality and reliability[18,19].

In EBM, accelerated electrons bombard the pre-sintered powder bed, converting their kinetic energy into heat and causing localized melting[20]. This interaction generates a melt pool. As the melt pool advances, the loose pre-sintered powder bed will be melted and then solidified to form a layer of the part. During melting and solidification, various defects can form[21–26]. Porosity is a primary concern because it directly compromises mechanical performance[27,28]. It can originate from gas trapped within powders, pore capture by the advancing solidification front, or lack-of-fusion between successive tracks and layers caused by insufficient energy input[29–32]. Surface irregularity is another major issue; its progressive accumulation across layers can reduce the printability and uniformity of parts[9,33]. Such irregularities can result from powder bed nonuniformity, adherence of partially melted particles or droplet spatter[18,34], and balling[35,36].

The defect formation mechanisms reported for EBM have all been observed in laser powder bed fusion (LPBF)[37–42]. However, EBM fundamentally differs from LPBF in its heat source: photons in LPBF penetrate only nanometers, causing 2D surface heating[20], whereas high-energy electrons in EBM can penetrate tens of microns, volumetrically depositing energy through scattering[43,44]. This fundamental difference raises a critical question: are there defect formation mechanisms and process instabilities unique to EBM?

[1]Department of Mechanical Engineering, University of Wisconsin-Madison, Madison, WI, USA. [2]Department of Materials Science and Engineering, University of Wisconsin-Madison, Madison, WI, USA. [3]X-ray Science Division, Advanced Photon Source, Argonne National Laboratory, Lemont, IL, USA. ✉e-mail: lianyi.chen@wisc.edu

To address this question, we developed an EBM system for in-situ synchrotron X-ray imaging and demonstrated its feasibility[32,45]. In this study, taking advantage of this in-situ characterization system, we discovered that bubble explosions during EBM in Al6061, unique to EBM, destabilize the melt pool and contribute to multiple types of defects. The findings provide critical mechanistic insights into the origin of melt pool instabilities and defect formation in EBM, laying the foundation for developing strategies to achieve defect-lean EBM additive manufacturing.

## Results

### Subsurface bubble formation and explosion

The dynamics of the melt pool and vapor depression in electron beam melting of Al6061 were captured using a custom-built electron beam melting system integrated with high-speed synchrotron X-ray imaging at beamline 32-ID of the Advanced Photon Source (Fig. 1a and details of the setup can be found in "Methods"). During melting, the melt pool remained confined within the boundaries of the Al6061 sample, ensuring the formation of a fully developed melt pool without penetrating the 1.2 mm sample thickness (a representative surface profile of the melt track is shown in Fig. 1b). Subsurface bubble formation and explosion were consistently observed in the electron beam irradiated area across all processing conditions, including stationary melting, shallow and deep vapor depression scanning (Fig. 2, Supplementary Movies 1–3), and were found to have impact on the melt pool and vapor depression dynamics.

### Dynamics of melt pool and vapor depression

The dynamics of the melt pool and vapor depression under the influence of subsurface bubble formation and explosion were investigated. Three distinct stages were identified during stationary melting: (i) initial formation of a melt pool (Fig. 3a–c), (ii) subsurface bubble formation and explosion (Fig. 3d–j), and (iii) periodic keyhole oscillation (Fig. 3j–o, Supplementary Note 1, Supplementary Fig. 1 and Supplementary Movie 4). The melt pool and vapor depression dimensions were tracked frame-by-frame over time (Fig. 3p). The moments corresponding to the transient states of melt pool and vapor depression shown in Fig. 3a–o are indicated by arrows along the curves for reference.

Stage 1 is characterized by the gradual formation and deepening of the melt pool without the appearance of a vapor depression. Stage 2 begins with the onset of rapid explosions, exceeding 10 kHz in frequency between 720 μs to 800 μs. Due to the short lifespan of individual bubbles and limitations in imaging frame rate, the exact frequency was not determined. At this stage, the rapid expansion of the melt pool and keyhole is primarily driven by rising vapor pressure within the depression, which displaces molten material, similar to the mechanism of keyhole drilling observed in laser powder bed fusion[46]. Bubble explosions may further amplify local pressure fluctuations, promoting additional keyhole deepening. Stage 3 marks the onset of periodic keyhole oscillation: while the width remains quasi-stable with fluctuations of approximately 20% of the mean width, the depth undergoes pronounced oscillations exceeding 60% of the mean depth, driven by periodic keyhole collapse at a frequency of approximately 2.5 kHz.

After the initiation of the melt pool and vapor depression, their evolution during scanning was investigated. To reach the specified power level at the start of the scan track, at the starting point of the scan track, the system was programmed to perform stationary melting for approximately 0.5 ms before initiating scanning. Once a vapor depression formed, the electron beam began scanning.

With a high scan speed of 1.2 m/s and an electron beam power of 426 W, frequent bubble explosions were observed at the front wall of the vapor depression, with multiple explosions occurring simultaneously within a single frame (Fig. 4, Supplementary Movie 5). These explosions generated surface waves at the vapor–liquid interface (Fig. 4j). The resulting unsteady melt pool dynamics manifest as a wavy morphology on the rear wall of the vapor depression, indicating a periodic and discontinuous driving force acting on the melt flow. Liquid metal accumulated at the rear of the melt pool, producing periodic humping and a rough melt track surface after solidification (Fig. 4i). The frequency of bubble explosion was quantified to exceed 24 kHz. Based on the number of explosions captured per frame, the front wall of the vapor depression was categorized into three distinct states: no explosion, single explosion (Fig. 4j), and multiple explosions (Fig. 4k), as summarized in Fig. 4l.

To further investigate the influence of energy density on bubble explosion dynamics and its consequences, electron beam melting was performed at a higher energy density by reducing the scan speed from 1.2 m/s to 0.6 m/s while maintaining the same electron beam power of 426 W. Similar explosion events were observed on the front wall of the vapor depression, along with explosion-induced surface waves at the vapor–liquid interface (Fig. 5, Supplementary Movie 6). A wavy morphology was also observed on the rear wall of the vapor depression

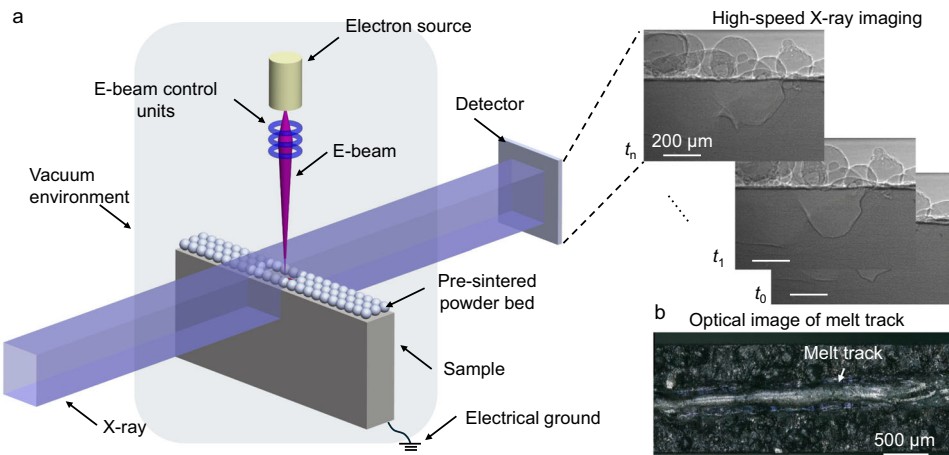

**Fig. 1 | In-situ electron beam melting high-speed synchrotron X-ray imaging experiment. a** An illustration of the experimental setup of in-situ high-speed synchrotron X-ray imaging of electron beam melting. **b** Typical surface profile of the melt track. High-speed X-ray imaging with a frame rate of 50 kHz, 100 kHz and 135 kHz, and spatial resolution of 2 μm was used to capture the dynamics of vapor depression, melt pool and spatter during electron beam melting. The experiments were conducted in a vacuum chamber using a beam with a diameter of approximately 200 μm (full width at half maximum (FWHM) of a Gaussian energy distribution). E-beam stands for electron beam.

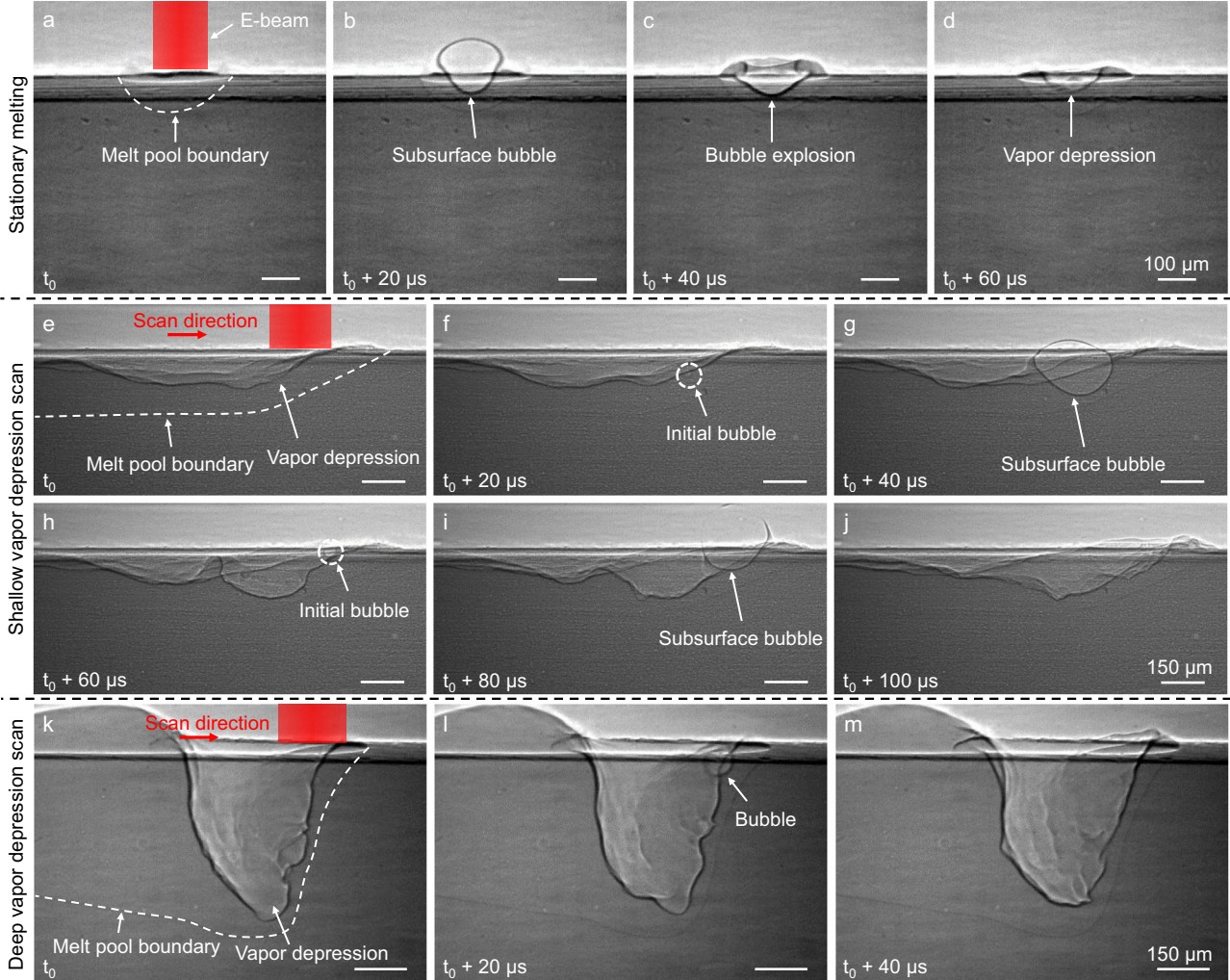

**Fig. 2 | Subsurface bubble formation and explosion in electron beam melting.** **a–d** X-ray images showing the dynamics of subsurface bubble formation and explosion during electron beam stationary melting with an electron beam power of 109 W. **e–j** X-ray images showing the dynamics of subsurface bubble formation and explosion during electron beam scanning under a shallow vapor depression at

318 W electron beam power and 1 m/s scan speed. **k–m** X-ray images showing the dynamics of subsurface bubble formation and explosion during electron beam scanning under a deep depression at 426 W electron beam power and 0.7 m/s scan speed. The electron beam is explicitly shown only in the first column; however, it remains turned on in all images.

(Fig. 5a–f). Following the onset of scanning, the melt pool and vapor depression gradually elongated (Fig. 5c, d). As the scan continued, molten metal accumulated at the rear of the melt pool. After solidification, the melt track exhibited a nonuniform height profile, with large humping. The classification of front wall states is shown in Fig. 5j. The total explosion frequency exceeds 31 kHz, with simultaneous multiple explosions occurring within a single frame at a frequency of approximately 13 kHz. This frequency is notably higher than that observed under lower energy density (Fig. 4l). Multiple explosions were consistently preceded by single-explosion events, suggesting a potential cascade or triggering mechanism in the explosion dynamics.

**Dynamics of spattering**
The dynamics of spattering under the influence of subsurface bubble formation and explosion were also investigated. Since spatter primarily arises from interactions between metallic vapor and the melt pool surface or powder bed, pre-sintered powder bed samples were used to study interactions among metal vapor, melt pool and powder. Following subsurface bubble explosions (Fig. 6a, b), the melt pool became unstable within the subsequent frames, ejecting a liquid ligament from the explosion site, followed by spatters (Fig. 6c–f, Supplementary Movie 7). To facilitate clearer identification of bubble

explosions and the associated spatters, the original images were processed and are shown in Fig. 6g, h. An additional example of bubble-explosion–induced spattering is provided in Supplementary Note 2, Supplementary Fig. 2 and Supplementary Movie 8. It is noted that strong evaporation at the vapor depression without bubble explosion can also drive spatter ejection, as previously reported in both laser and electron beam powder bed fusion processes[34,47]. The bubble explosions observed here represent an additional mechanism that can contribute to spatter formation.

## Discussion
### Subsurface heating induced subsurface bubble formation and explosion
Upon electron beam activation, the high-energy electron beam begins to strike the metal surface. The interaction between the electron beam and metal can be understood as the interaction between incoming high-speed high-energy electrons and the atomic nuclei and valence electrons in metallic samples. In metals, valence electrons hold positively charged metal ions in their equilibrium positions, maintaining a solid state[48]. When incoming electrons interact with metallic atomic nuclei via Coulomb forces, their trajectories are deflected, but energy loss remains minimal[49]. In contrast, inelastic collisions with valence

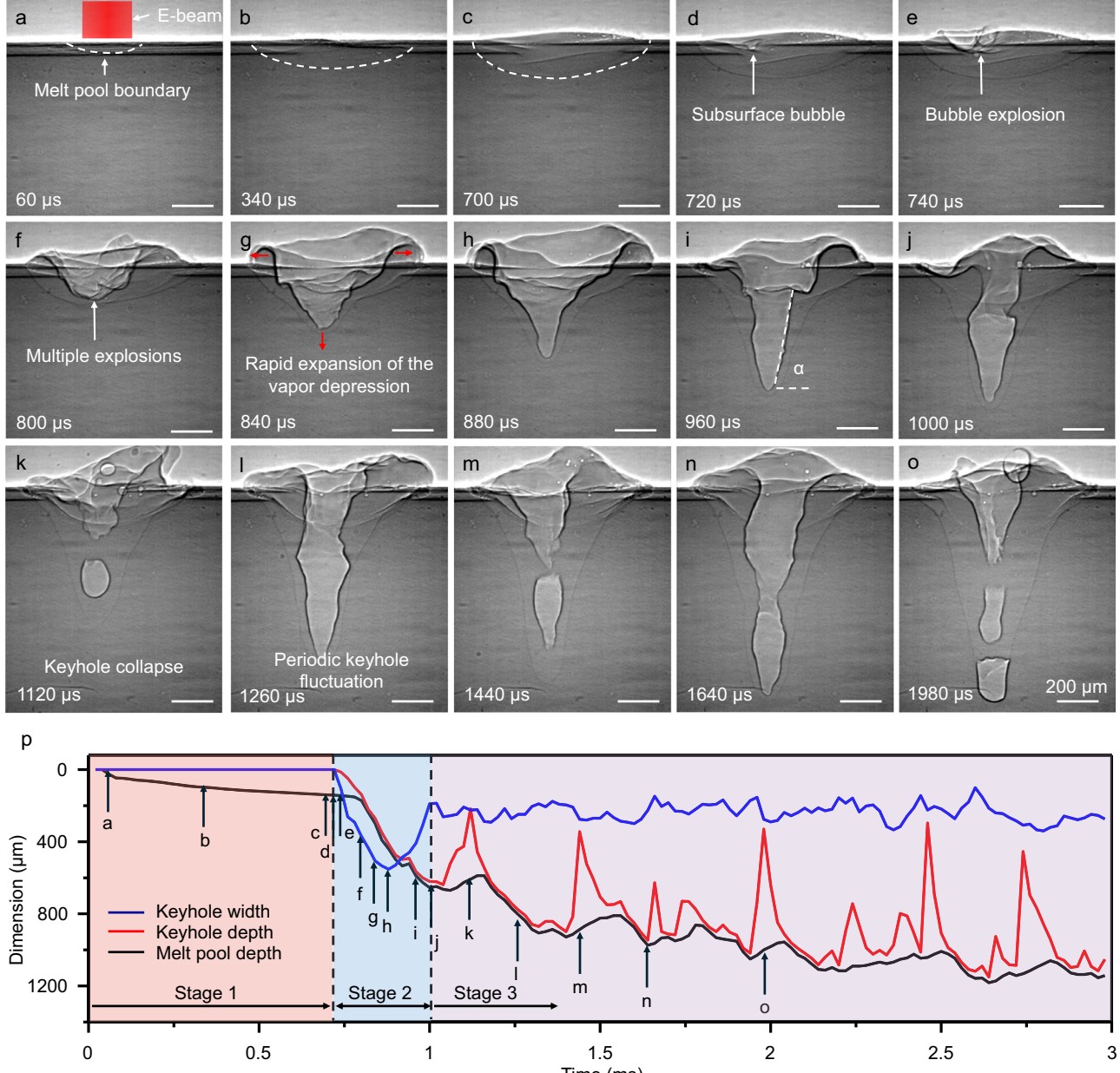

**Fig. 3 | Initiation of keyhole and melt pool under the influence of subsurface explosion during electron beam stationary melting with an electron beam power of 426 W. a–c** Stage 1: Initial formation of a melt pool during electron beam stationary melting. **d–j** Stage 2: Subsurface bubble formation and explosion. At this stage, violent bubble explosions can contribute to the rapid deepening and widening of the keyhole. **j–o** Stage 3: Periodic keyhole oscillation. In this stage, the keyhole depth starts to fluctuate periodically. **p** Dimensions of melt pool and keyhole over time. Source data are provided as a Source Data file.

electrons dominate the energy transfer process, resulting in significant energy deposition[50,51]. A portion of this energy is converted into thermal excitation, generating lattice vibrations. When these vibrations become sufficiently intense, metallic bonds break, freeing atoms from their equilibrium positions and initiating melting. This process ultimately leads to the formation of a melt pool on the metal surface in Stage 1 (Fig. 3a–c).

Upon entering the material, electrons possess high initial velocities, resulting in brief interaction times with valence electrons. Consequently, the inelastic scattering cross-section is initially small, and the energy deposition rate remains low. As electrons undergo successive scattering events and gradually lose velocity, their energy deposition per unit path length increases. This leads to a peak in energy deposition at a specific depth[43,44], beyond which the electrons lose their remaining energy, and deposition rate decreases, as shown

in Fig. 7. For aluminum irradiated with a 60 keV electron beam, the peak energy deposition depth ($d$) is approximately 10 μm beneath the surface[44].

Thus, as heating progresses, the temperature at the maximum energy deposition depth may reach the boiling point first. Simulation results by Yan et al. support this hypothesis, showing that under 60 keV electron-beam irradiation of Ti-6Al-4V, the subsurface region (~4.4 μm below the surface) reaches the highest temperature first, on the order of 3900–6000 K[43]. This localized overheating could initiate vaporization and the nucleation of subsurface bubbles filled with metal vapor (Fig. 3d). Future studies will focus on investigating the mechanisms underlying bubble formation. Continued energy absorption increases the vapor volume and raises its temperature, causing the bubble to expand (Fig. 3e). Additionally, the presence of a vacuum environment further accelerates bubble expansion. As the bubble

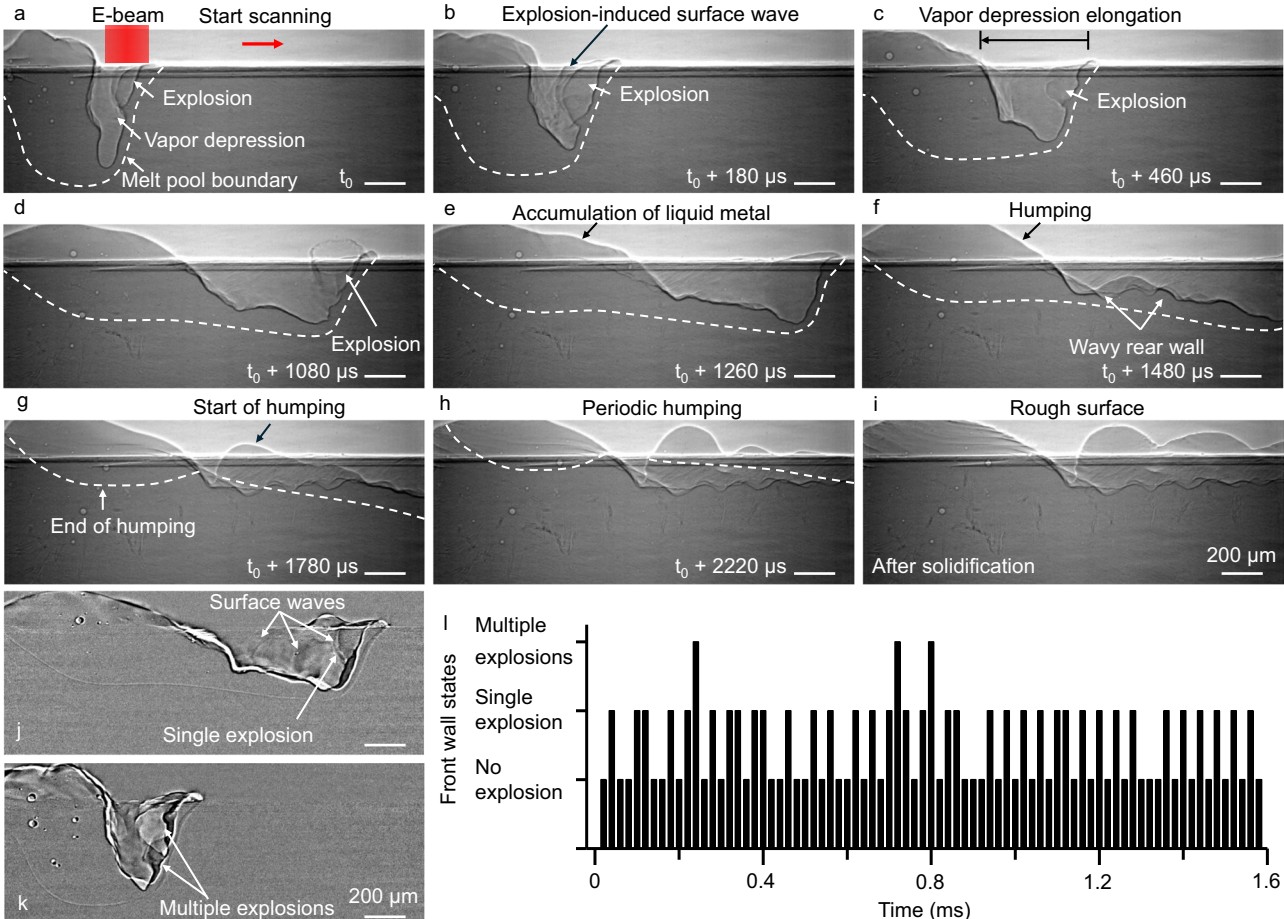

**Fig. 4 | Evolution of the vapor depression and melt pool under the influence of subsurface explosion during electron beam scanning with a high scan speed of 1.2 m/s. a–h** X-ray images showing the evolution of the vapor depression and melt pool under the influence of subsurface explosion during electron beam scanning with an electron beam power of 426 W and a scan speed of 1.2 m/s. **i** X-ray image showing the nonuniform melt track after scanning. **j**, **k** Processed X-ray images showing single explosion and multiple explosions during melting. **l** Front vapor depression wall states over time. Source data are provided as a Source Data file.

grows and approaches the liquid surface, the upper liquid film gradually thins due to downward liquid drainage[52,53]. Eventually, the film becomes unstable and ruptures, resulting in a bubble explosion.

## Evolution of the melt pool and vapor depression induced by bubble explosion

Upon bubble explosion, the high-pressure high-temperature vapor inside the bubble is rapidly released, converting stored energy into an explosive event. The thin liquid film covering the bubble ruptures violently, and the molten metal around the explosion is blown out into spatters (Fig. 6). Previous research has shown that when the electron beam interacts with powder particles above the powder bed, these particles can become charged and dispersed beyond the focal spot before eventually falling back onto the powder bed, thereby triggering smoking[54]. Similarly, spatters produced by subsurface bubble explosions may interact with the electron beam, become charged, and consequently act as potential initiators of smoking events.

Simultaneously, after the bubble explosion, vapor expands into the surrounding space, generating pressure waves (Figs. 4, 5). These dynamics contribute to reshaping the melt pool and vapor depression through two coupled mechanisms: (1) impulsive liquid displacement driven by vapor expansion and (2) modulation of energy absorption behavior due to geometric evolution of the vapor depression.

The first mechanism, impulsive liquid displacement, arises from the high-frequency bubble nucleation and explosion in Stage 2. Each bubble explosion generates localized pressure, and the cumulative effect of multiple bubble explosions results in overlapping pressure fields, which forcibly displace liquid metal radially outward and accelerate keyhole drilling. The expansion of keyhole walls exhibits roughness and wrinkling, indicating the intense pressure exerted by successive explosion events (Fig. 3f–h).

The second mechanism, modulation of energy absorption behavior, results from the geometric evolution of vapor depression. As the vapor depression deepens, the incidence angle ($\alpha$) at the vapor-liquid interface approaches ~82° (Fig. 3i), significantly altering the electron energy deposition profile. The effective peak energy deposition depth ($d_0$) decreases with increasing $\alpha$, following $d_0 = d \cdot \cos \alpha$ (Fig. 7b, c). The effective peak energy deposition depth is defined as the distance from the vapor–liquid interface to the layer of peak energy deposition measured along the direction normal to the liquid–gas interface (Fig. 7b). At $\alpha = 82°$, $d_0$ is reduced to ~1.4 μm, (~14% of $d$), concentrating energy deposition near the surface rather than deeper regions. Additionally, the formation of a deep keyhole facilitates a more uniform distribution of electron beam energy along the keyhole walls, diminishing localized energy concentration. Together, these effects mitigate subsurface bubble nucleation and weaken the impact of bubble explosions. As subsurface bubble nucleation and explosion diminish, surface tension drives a contraction in vapor depression width. Consequently, the keyhole width expands sharply in the first half of Stage 2 due to intense bubble explosions and contracts in the latter half as explosions subside.

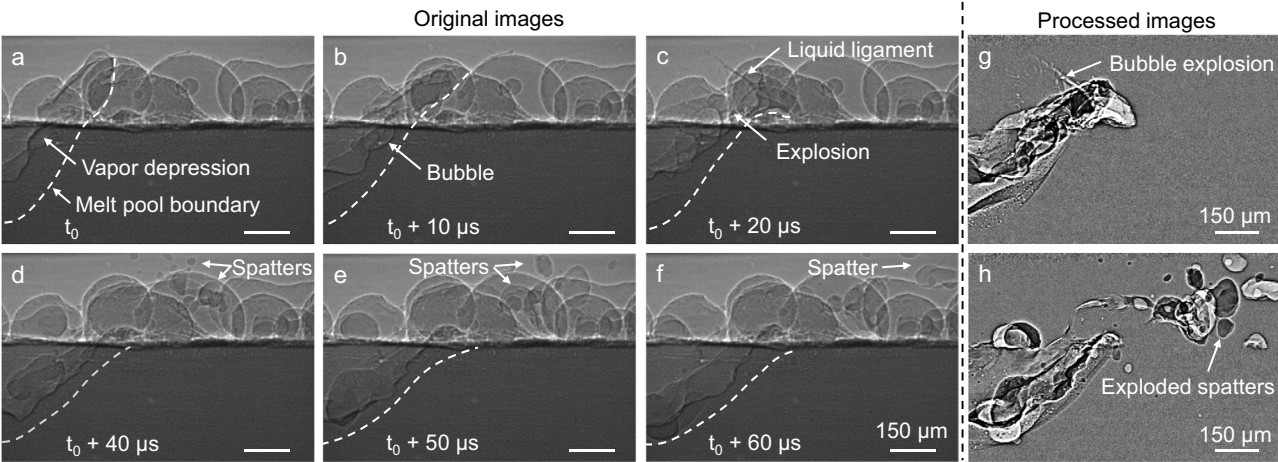

**Fig. 5 | Evolution of the vapor depression and melt pool under the influence of subsurface explosion during electron beam scanning with a low scan speed of 0.6 m/s. a–h** X-ray images showing the evolution of the vapor depression and melt pool under the influence of subsurface explosion during electron beam scanning with an electron beam power of 426 W and a scan speed of 0.6 m/s. **i** X-ray image showing the nonuniform melt track with large humping after scanning. **j** Front vapor depression wall states over time. Source data are provided as a Source Data file.

**Fig. 6 | Dynamics of spattering under the influence of subsurface explosion during electron beam scanning with an electron beam power of 385 W and a scan speed of 1.0 m/s. a–f** Original X-ray images showing the dynamics of spattering induced by bubble explosion during electron beam scanning. **g** Processed X-ray image showing the bubble explosion. **h** Processed X-ray image showing exploded spatters induced by bubble explosion. Image **g** was obtained by dividing the intensity at each pixel in image **b** by the intensity of the corresponding pixel in image **c**. Similarly, image **h** was derived by dividing the intensity at each pixel in image **e** by the intensity of the corresponding pixel in image **f**.

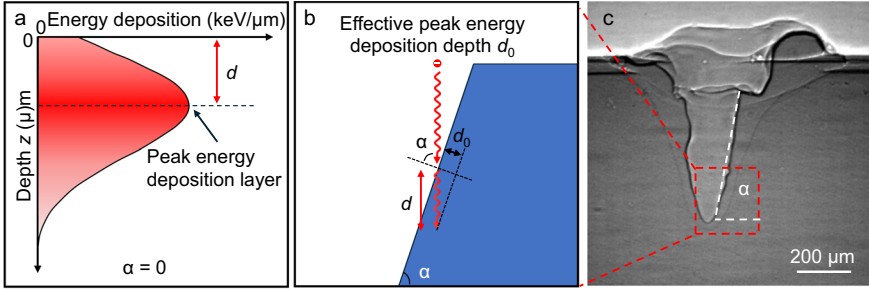

**Fig. 7 | Energy deposition profile and peak energy deposition depth under electron beam irradiation. a** A schematic illustration of energy deposition profile with an electron beam incident normal to the surface (inclination angle $\alpha = 0$). **b** Schematic showing the effective peak energy deposition depth when the electron beam strikes the inclined keyhole wall (inclination angle $\alpha > 0$). **c** A representative X-ray image captured during electron beam melting, showing the formation of a deep keyhole.

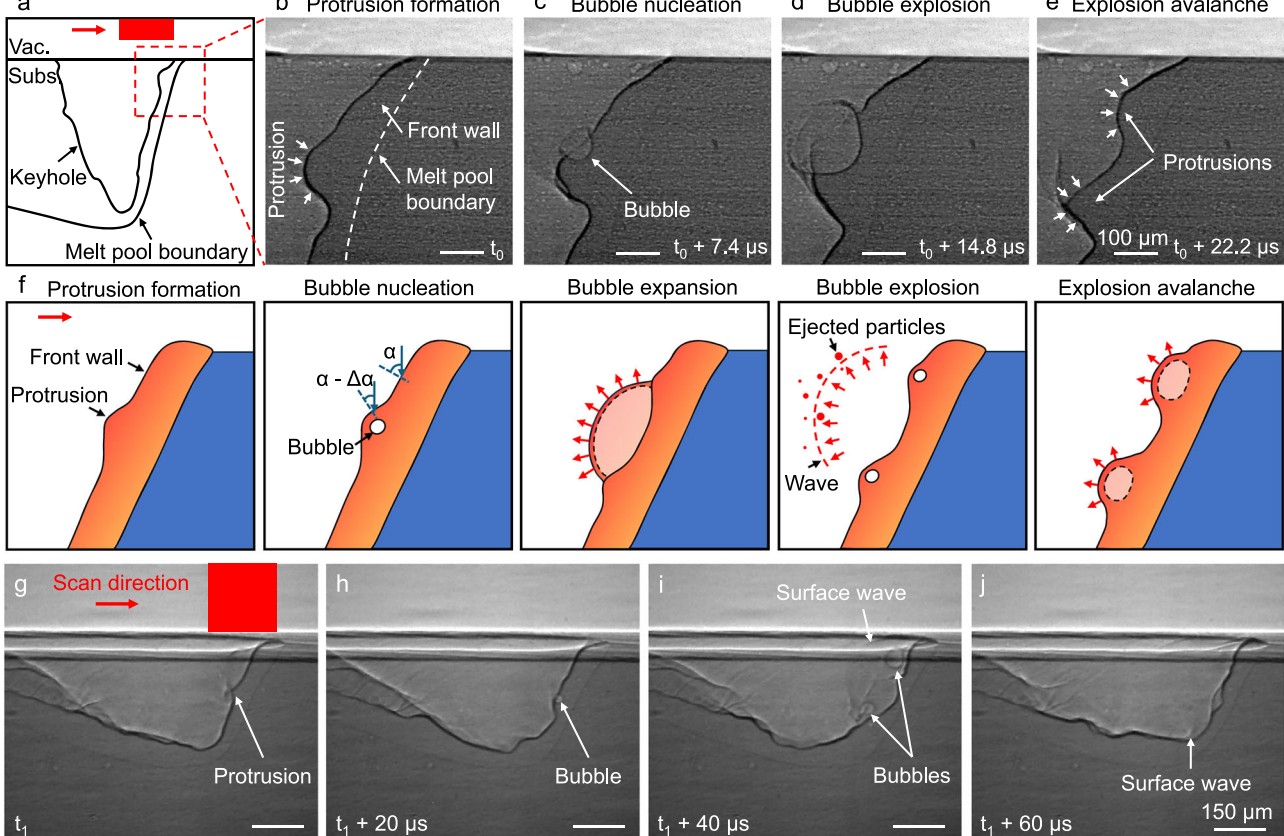

**Fig. 8 | Mechanisms of subsurface bubble formation, explosion and avalanche during electron beam scanning. a** A schematic diagram showing the typical morphology of the keyhole and melt pool during electron beam scanning. **b–e** High frame rate X-ray images showing the dynamics of subsurface bubble formation and explosion on the front keyhole wall during electron beam scanning with an electron beam power of 318 W and a scan speed of 0.3 m/s. **f** A schematic diagram showing the mechanisms of subsurface bubble formation, explosion and avalanche in electron beam scanning. **g–j** X-ray images showing a similar sequence of bubble formation, explosion and avalanche during electron beam scanning with an electron beam power of 318 W and a scan speed of 0.7 m/s. Vac. stands for vacuum, and subs. stands for substrate.

During melting, recoil pressure from the dense metallic vapor drives the surrounding liquid outward, forming a deep and narrow cavity (Fig. 3i, j). As melting continues, the interplay among thermocapillary force, Marangoni convection, recoil pressure, and surface tension, similar to the mechanisms observed in laser powder bed fusion[55,56], induces keyhole instability. When the keyhole collapses, the high-energy electron beam reinitiates intense evaporation in the newly formed shallow keyhole, driving a cyclic collapse–regrowth instability (Fig. 3k–o).

## Mechanisms of bubble explosion avalanche during scanning

To investigate the dynamics of cascading bubble explosions at the front wall of the vapor depression during scanning, high frame rate in-situ imaging of electron beam scanning was conducted with a magnified view of the front wall region. The imaging region relative to the entire melting area is shown in Fig. 8a.

On the inclined front wall of the vapor depression, instabilities induced by bubble explosions lead to the formation of protrusions (Fig. 8b). Once a protrusion forms, its upper surface presents a

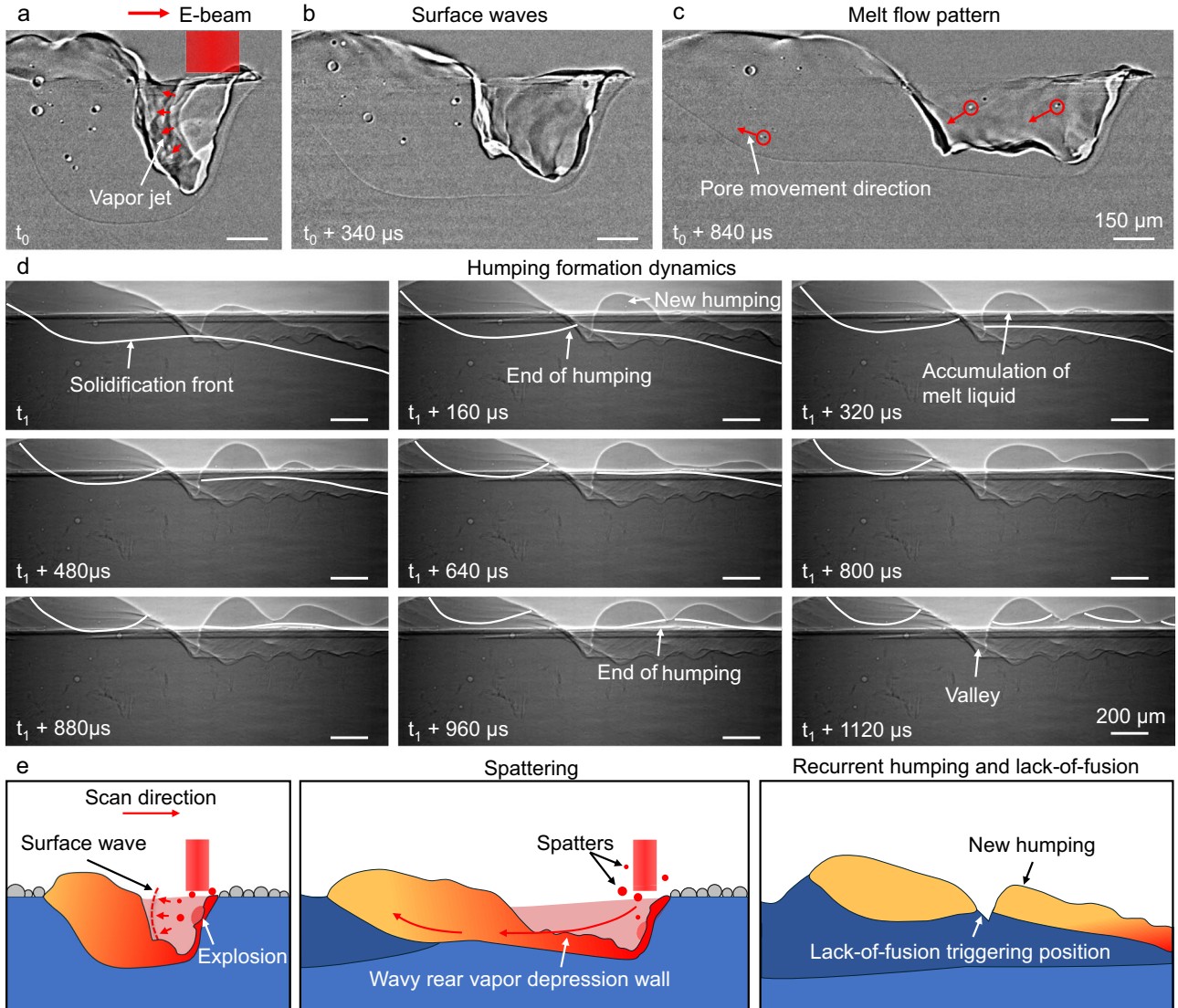

**Fig. 9 | Mechanisms of the influence of bubble explosion on melt pool instabilities and defect formation during scanning. a–c** Processed X-ray images showing the dynamics of the melt pool and vapor depression under the influence of bubble explosion on the front keyhole wall in electron beam scanning. **d** X-ray images showing the dynamics of periodic humping formation during electron beam scanning with an electron beam power of 426 W and a scan speed of 1.2 m/s. **e** A schematic diagram showing the mechanisms of melt pool instabilities and defect formation in electron beam scanning.

reduced local inclination angle ($\alpha - \Delta\alpha$, indicated in Fig. 8f), allowing the electron beam to penetrate deeper into the protrusion. The region beneath the protrusion surface reaches the boiling point of aluminum more rapidly at the location of maximum energy absorption, initiating bubble nucleation (Fig. 8c). Continued energy absorption drives bubble expansion and eventual explosion, ejecting the thin overlying liquid film as droplets into the vacuum (Fig. 8d).

Simultaneously, during bubble expansion and explosion, displaced molten metal accumulates around the bubble perimeter, forming new protrusions (Fig. 8e, Supplementary Movie 9). New bubbles can be initiated within the freshly formed protrusions. This leads to a cascade of sequential explosions, a phenomenon described as an explosion avalanche. This mechanism explains the observation that multiple explosions are preceded by a single explosion event at high energy density. A schematic illustration of the mechanisms of bubble formation, expansion and avalanche during scanning is presented in Fig. 8f. A similar sequence of bubble formation, explosion and avalanche is shown in Fig.8g–j under a lower energy density condition, using the same electron beam power of 318 W while increasing the scan speed from 0.3 m/s to 0.7 m/s (Supplementary Movie 10).

## Impact of bubble explosion on melt pool instabilities and defect formation

During scanning, explosions at the front wall can directly influence the melt pool and vapor depression dynamics through two primary effects. First, vapor jets from explosions impinge on the rear wall, exerting dynamic pressure that drives the horizontal extension of the vapor depression. The high frequency of explosions results in overlapping vapor jets and superposed pressure waves, amplifying local pressure and enhancing vapor depression elongation (Fig. 9a, b). Second, the backward melt flow observed in Fig. 9c, a phenomenon commonly reported in laser powder bed fusion, typically arises as molten metal circulates around the vapor depression and gains momentum opposite to the scanning direction[57]. Bubble explosions occurring at the front wall may introduce additional momentum to the backward flow by generating transient vapor jets and surface waves that propagate along the vapor–liquid interface.

As the melt flow propagates backward, periodic undulations develop along the melt pool surface due to the Rayleigh–Taylor instability, as previously reported in laser and electron beam powder bed fusion[34,41]. As the solidification front advances, these undulations

evolve into humping structures along the melt track. In electron beam melting, intermittent momentum impulses generated by bubble explosions further amplify this instability by perturbing the melt pool surface. The combined effects of Rayleigh–Taylor and explosion-induced instabilities thus promote periodic undulations, ultimately leading to the formation of humping structures along the melt track (Fig. 9d, Supplementary Movie 5). Deep valleys form between adjacent humps, and during subsequent layer deposition, these valleys are difficult to fully melt. As a result, lack-of-fusion porosity can be initiated at these sites, making them critical lack-of-fusion porosity-triggering locations. A schematic illustration of the influence of bubble explosion on melt pool instabilities and defect formation during electron beam scanning is presented in Fig. 9e.

The influence of preheating on melting dynamics, including subsurface bubble formation, melt pool behavior, and vapor depression evolution, was also investigated. Subsurface bubble formation and explosions were observed, accompanied by similar melt pool dynamics (Supplementary Note 3, Supplementary Fig. 3). To suppress melt pool instabilities induced by bubble explosion, several strategies can be explored. First, optimizing alloy composition by reducing the concentration of volatile elements may enhance melt pool stability. Second, applying energy modulation techniques, including pulsed beam operation, may prevent extreme peak explosion by temporally distributing the energy input. Furthermore, the effects of electron beam acceleration voltage on bubble explosion behavior and melt pool dynamics warrant systematic investigation.

In summary, we discovered that bubble explosion is a new mechanism for melt pool instability and defect formation in the electron beam melting additive manufacturing process. The bubble explosion is induced by the unique energy absorption profile of the electron beam, i.e., the maximum energy absorption occurs beneath the surface of the material. The mechanism elucidated here could inspire and guide the development of EBM machines, processing strategies, and alloy compositions for achieving additive manufacturing of defect-lean components for mission-critical applications. This work is expected to inspire the electron beam additive manufacturing and welding community to advance modeling efforts for a quantitative understanding of bubble formation, explosion, and their effects, and to experimentally investigate, through in-situ monitoring, the occurrence of subsurface bubble explosions in high-atomic-number materials, thereby deepening the overall understanding of electron beam melting.

## Methods
### Materials
Al6061 was selected as the material for this study because of its good X-ray transparency for in-situ high-speed X-ray imaging experiments[58]. Two types of samples were prepared for the experiments: bare substrates and pre-sintered powder-bed samples. The bare substrate samples were sectioned from commercial Al6061 plates (purchased from McMaster). Since powder bed sintering is not the focus of this study, to accelerate the beamline experiments, the powder layer was pre-sintered before loading it into the chamber. The pre-sintered powder-bed Al6061 samples were prepared by sintering one layer of Al6061 powders (15–45 μm) on a commercial Al6061 plate using laser heating under an argon atmosphere to minimize oxidation. As observed in the X-ray images, the powder particles appear to have partially melted and coalesced into larger agglomerates, with apparent diameters of approximately 200 μm. This coalescence results from localized laser melting during the sintering process rather than the use of larger initial powder particles. Both types of Al6061 samples were cut into pieces with dimensions of 58 mm (length) x 10 mm (width) x 1.2 mm (thickness in the X-ray penetration direction), with the powdered surface facing the incident beam during experiments.

### Electron beam melting system
High-speed, high-resolution X-ray imaging was conducted using a custom-built electron beam melting system integrated with beamline 32-ID-B at the Advanced Photon Source, Argonne National Laboratory (Lemont, IL). The system comprises a vacuum chamber with multiple customized viewports, an EBM unit, and a sample stage. The EBM gun operates in both continuous mode (max power 600 W) and pulsed mode (peak power up to 1.2 kW). It features a LaB₆ single-crystal cathode, magnetostatic beam-shaping /focusing /deflection coils, and dual turbopumps. The system offers a maximum emission current of 10 mA and a tunable accelerating voltage range of 0–60 keV, enabling both single-track and stationary melting experiments on pre-sintered powder beds and bare substrates. In the experiments, the electron beam has a diameter of approximately 200 μm at the working plane, defined as the full width at half maximum (FWHM) of a Gaussian energy distribution. The beam size was measured using a slit-scan method, in which the beam was stepped across a slit of known width while measuring the transmitted current to determine the rise/fall regions. Then the spot size was calculated using the FWHM metric. The vacuum system achieves a base pressure of ~$10^{-8}$ torr using copper-gasket seals and features glassy-carbon X-ray viewports (6 mm thick) that maintain both high vacuum integrity and high X-ray transparency. All experiments were performed under high vacuum (~$10^{-5}$ torr), without helium shielding gas.

### In-situ high-speed synchrotron X-ray imaging
The X-ray imaging system at beamline 32-ID utilized a pseudo pink beam generated by an 18-mm-period undulator, with a single harmonic energy peak centered at approximately 24 keV with a bandwidth of 5–7%. The beam was transmitted through the sample and converted to visible light via a LuAG:Ce scintillator (100 μm thick), then captured by a high-speed camera (Photron FastCam SA-Z2100K). The imaging setup achieved a nominal spatial resolution of 2 μm per pixel, with recording frame rates of 50 kHz, 100 kHz and 135 kHz.

### Surface morphology characterization
Surface profiles of single scan tracks were characterized using a VHX-5000 Digital Microscope (KEYENCE Corporation of America) equipped with a fringe-projection optical system. This technique enables full-field, three-dimensional mapping of the surface topography of individual tracks.

### Image processing
ImageJ 1.53t was employed for enhancing the contrast of the melt pool surface and keyhole boundary, and quantifying melt pool and vapor depression dimensions[59]. The depths of the melt pool and vapor depression were measured from the sample surface to the deepest point of the melt pool and vapor depression, respectively. And the vapor depression width was measured at the vapor depression opening. The apparent boundaries of the melt pool and keyhole in the X-ray images possess a width of 2 pixels (equivalent to 4 μm), resulting in an inherent uncertainty of ± 2 μm for all measurements. To clearly identify surface waves and bubble trajectory, the image intensity at each pixel of Frame (i + 1) was divided by the intensity of the corresponding pixel in Frame (i), converting motionless parts in the image to a blank background[60]. More details about image processing can be found in the Supplementary Information (Supplementary Note 4, Supplementary Fig. 4).

To quantify bubble explosion events, a customized detection and classification protocol was employed. Explosions were identified using two complementary criteria applied to sequential frames: (1) the sudden appearance of a localized, high-intensity bright region on the front wall of the vapor depression, indicative of vapor release and liquid-film rupture; and (2) associated frame-to-frame morphological changes at the same location, such as rapid expansion or distortion. Based on these criteria, the front-wall state in each frame was classified into

three categories: no explosion, defined by the absence of a newly formed bright region; single explosion, characterized by one spatially isolated bright-region cluster; and multiple explosions, indicated by two or more spatially distinct bright-region clusters within the same frame. The primary source of uncertainty in event counting arises from the limited temporal resolution relative to the explosion dynamics. Because the frame interval is limited to 20 μs (50 kHz), explosions with lifetimes shorter than the inter-frame time may be missed or ambiguously attributed to adjacent frames. Consequently, the measured explosion frequency should be regarded as a lower estimate of the actual rate. For each condition, the analysis is based on more than 200 data points extracted from high-speed imaging frames.

## Data availability
The authors declare that the data supporting the findings of this study are available within the paper and its supplementary information files Source data are provided with this paper.

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

## Acknowledgments

This work is supported by Vilas Associate Award (L.C.) and the U.S. Department of Commerce (Award ID number: 70NANB21H039, L.C.). This research used resources of the Advanced Photon Source, a U.S. Department of Energy (DOE) Office of Science user facility operated for the DOE Office of Science by Argonne National Laboratory under Contract No. DE-AC02-06CH11357. The authors would like to thank Drs. Fan Zhang, Lyle Levine, Brandon Lane, Brian Simonds, and Nik Hrabe from the National Institute of Standards and Technology (NIST) for fruitful discussions. The authors appreciate Drs. Scott Sanders, Matthias Beuting, Brandon Walker and Kevin Eliceiri at the University of Wisconsin-Madison for their helpful suggestions.

## Author contributions

L.C. and J.Y. conceived the idea. L.C. supervised the research project. J.Y. and L.C. designed the experiment. L.I.E. S.J.C., K.F., J.Y., J.H., A.N., Q.L., and L.C. conducted the X-ray imaging experiments. J.Y. performed the data analysis and characterization. J.Y. and L.C. wrote the paper with input from all authors. All authors discussed the results and reviewed the manuscript.

## Competing interests

The authors declare no competing interests.
