## [Transparent Peer Review file · Nature Communications]

Bubble explosion induced melt pool instabilities in electron beam melting of aluminum alloy

Corresponding Author: Professor Lianyi Chen

Version 0:

Reviewer comments:

Reviewer #1

(Remarks to the Author)

This manuscript used in-situ high-speed high-resolution synchrotron X-ray imaging technology to observe the formation and explosion of subsurface bubbles in the molten pool during the electron beam powder bed fusion (EB-PBF) process of aluminum alloy Al6061, and investigated the mechanism of the subsurface bubble formation and the effects on the molten pool instabilities, and the morphology of the melt track. Since it is the first time that subsurface bubbles have been directly observed during the melting and deposition process of EB-PBF, this manuscript is of great significance for the research of EB-PBF technology.

Although the classical EB process theory suggested that the energy absorption peak of the EB is several to tens of micrometers below the surface of the material, there is no experimental observation directly indicating the formation of micro-bubbles and its explosions below the surface. Moreover, for the aluminum alloy used in this paper as the experimental material, the frequency of bubble formation and explosion is very high, almost existing under all energy density conditions. Therefore, the findings of this paper are original.

This manuscript is valuable and an important reference for understanding the EB-PBF process, but more evidence is needed to prove that its conclusions are universal and can guide the process development for other materials. The main reason is that the aluminum alloy used in this experiment is an uncommon material for EB-PBF process with low density, low melting point and boiling point, which makes the subsurface bubble easy to form. On the contrary, the common materials for EB-PBF, such as titanium alloys, titanium aluminum alloys and superalloys, have higher melting points and boiling points, and higher difficulty to form the bubbles. So, more evidence is required to prove the existence of bubble formation and explosion phenomena in these materials as well. Here, the factors need to be considered not only the higher melting and boiling points of the material, but also the higher material density, which makes the position of the EB energy absorption peak closer to the surface. Therefore, more experimental research may be needed to verify whether the bubble formation and explosions in EB-PBF are "not material-specific and is expected to manifest across all material systems" (Page 15, line 308). Before the author provides more evidence, it is the best for the title and conclusion of this manuscript to focus on the EB-PBF process for aluminum alloys.

Another point worth discussing is, the manuscript mentioned in Page 16 line 314, the strategy of an appropriate EB accelerating voltage to shift the peak energy deposition closer to the surface to suppress melt pool instabilities induced by bubble explosion. Actually, shifting the energy deposition closer to the surface usually needs to lower the acceleration voltage of the EB. But existing research (Ref. 1) has shown that although increasing the EB acceleration voltage will deepen the position of energy absorption, it can also increase the total depth of EB energy absorption effectively, and making energy absorption more uniform at different depths. Meanwhile, keeping the constant power, the higher acceleration voltage will make the intensity of the EB low, which can further reduce the energy density of the EB energy absorption peak, and should be beneficial for reducing the bubble formation as well. Additionally, the exiting research (Ref. 1) indicated already that higher acceleration voltage of the EB could obtain higher density and fewer defects, which may be partially due to the reduction of bubble formation.

In addition to the above issues, the experimental methods, data analysis, interpretation and conclusions of the manuscript are quite appropriate.

Finally, one more suggestion that the authors may be interested. Based on the engineering practice, the formation and explosion of the subsurface bubbles may not only cause melt pool instabilities, lack-of-fusion defects, and melt track irregularity, but also may trigger the phenomenon of "smoking", which wasn't mentioned in the manuscript yet. Existing research (Ref. 2) has revealed that when EB encounters powder particles above the powder bed, it will charge the particles and be dispersed onto the powder bed outside the focal spot, and the charged particles will also fall down onto the powder

bed, thereby triggering smoking. The spatters, generated by the subsurface bubble explosions, may be a trigger to cause smoking as well if they can fly to a higher altitude and coincidentally encounter the EB. This may explain the sudden occurrence of smoking during selective melting with a focused EB in the EB-PBF process. If recognized by the authors, it can be added to the background introduction of the manuscript.

Reference:

1. "Effects of the higher accelerating voltage on electron beam powder-bed based additive manufacturing of Ti6Al4V alloy." Additive Manufacturing 50 (2022): 102579.
2. "Multiple stages of smoking phenomenon in electron beam powder bed fusion process." Additive Manufacturing 66 (2023): 103434.

Reviewer #2

(Remarks to the Author)

The authors report the electron beam additive manufacturing specific process instability caused by bubble explosion. Further, the observations are connected and explained by the electron beam matter interaction. The reported results provide important insights that will facilitate improved process control and final quality of manufactured components. Especially figure 7 and 8 provide sound evidence of the claims.

1. Further details on the prepared samples can be provided. How was the sintering performed? Could there be gas or humidity entrapped in the samples?
2. In line 42-43 the authors claim that boiling is always present in ebm processing. But steering the processing parameters allows for selection of keyhole or conduction mode melt pools. It is not necessary to process with keyhole.
3. The sentence 249-250 should be rephrased.
4. Line 308-309 is it also expected in tungsten?

Reviewer #3

(Remarks to the Author)

The authors present a highly novel and compelling study about a newly discovered bubble formation and explosion phenomenon that occurs during melt pool formation within Al6061 samples using EBM (better referred to as PBF-EB nowadays), enabled by high-speed synchrotron imaging with a specialized in-situ sample environment. The discovered phenomenon is novel and has never been reported on, making it highly suitable for publication. Unfortunately, the presentation of the phenomenon, as well as its discussion in its current state, is highly flawed and requires substantial revision. First of all, the lack of supplementary video material (which is often included with such in-situ studies) as well as appropriately processed imagery focusing on the phenomenon itself, make it highly difficult for the reader to confirm the claims and descriptions in the written paragraphs. The imagery should be improved for better clarity, focusing on the bubble explosion phenomenon and ideally video material should also be provided if possible. Next, the provided explanations for observed phenomena are highly superficial and lack sufficient theoretical and mathematical considerations to be convincing. Here, the authors seem to provide explanations that best suit the narrative of their article instead of considering the established complexity and nuance of melt pool dynamics in powder bed fusion additive manufacturing. Additionally, the authors attribute essentially all observed melt pool dynamics (keyhole shape, periodic collapse, melt pool backflow, spattering, etc.) to the bubble explosion phenomenon, despite there being an established body of scientific work (predominantly in-situ imaging of laser powder bed fusion, as well as CFD simulations) providing much more elaborate and convincing explanations for many of the described phenomena. The article, in its current state, is also entirely devoid of a discussion of observed phenomena that considers previously established descriptions of melt pool dynamics in powder bed fusion AM which seems highly inappropriate, as many of the observed phenomena very closely resemble the findings of other research groups and studies. Since the provided explanations for observed phenomena are mostly phrased as factual truths and not as working hypotheses, the authors are advised to provide a much more elaborate analysis of their experimental findings to support their claims, in order to demonstrate a consideration of established research in this field, especially considering the novelty of their discovered phenomenon. Alternatively, if this turns out to be outside the scope of this article, it should be made clear that the bubble explosion phenomenon could potentially be a contributing factor in most of the observed melt pool dynamics and not their sole cause. Considering the novelty of the presented findings, the article is definitely worthy of publication once these general concerns, as well as those described below have been addressed.

Abstract

I.19: The statement that bubble explosions are intrinsic to EBM is overselling the findings, as a single study with a single material is not sufficient evidence for the introduction of bubble explosions as an intrinsic mechanism of this AM technology.

Introduction

I.44: EBM can create vapor depressions under certain conditions but does not necessarily lead to their formation in general.
I.60: Similar to the Abstract, "intrinsic and unique to EBM" is overstating the significance of the presented findings.

Results

L.78: You do not provide a spot size definition for their stated electron beam spot size. Is the spot size 200 μm FWHM (Full Width Half Max), FWHP (Full Width Half Power), $D4\sigma$ (4 standard deviations) or some other spot size definition?

L.101: You state that the rapid melt pool and keyhole expansion are the result of bubble explosions, yet there is no reason for this being the case. The increase in both can easily be attributed to increasing vapor pressures which result in a displacement of the molten material and lead to an increase in heat flux into the still solid substrate.

L. 107-110: Again, there is no necessary causal link between the shown vapor depression formation and the effect of bubble formation and explosion. The widening of the depression at the beginning could simply be the result of material transport phenomena. The fact that keyholes get more narrow as they deepen (e.g. common in laser powder bed fusion) can be explained by the necessarily higher vapor pressures to maintain them. Since the electron beam most likely follows a gaussian distribution, evaporation is strongest in its center and thus the keyhole gets more narrow as it deepens as a consequence, as only the energy flux density in the beam's center is sufficient to displace the melt. While the bubble explosions could be a contributing factor, there is no compelling evidence to suggest that it dominates the melt pool dynamics. How many datasets did show the behavior presented in Fig. 3f – h, exactly? Is the shown example statistically significant? Have any calculations been performed to estimate the force exerted by bubble explosions onto the surrounding melt, compared to the force exerted by evaporation directly? Is there an underlying physical model that strongly suggests that bubble explosion could dominate melt pool displacement at all? If so, why was it not included in the manuscript?

L.110-113: Based on Fig. 3j, the onset of keyhole collapse seems to be attributable to the cessation of bubble explosions. Is this assumption correct? If so, be aware that keyhole collapse is a phenomenon not unique to EBM and does happen in PBF-LB as well where this bubble explosion behavior as never been observed. See the study of Qu et al.

(<https://doi.org/10.1016/j.addlet.2022.100068>) for reference. The periodic collapse can be explained by Marangoni convection induced material transport in the keyhole surrounding melt pool, as well as fluctuations in electron beam irradiation due to changes in beam incidence angles and the effect of surface tension on the shape of the keyhole.

Fig.4 & 5: The bubble formation and explosion events are extremely difficult to discern from the provided radiography data, so much so that the reader is forced to believe the described dynamics without being able to confirm them through the provided imagery. I highly suggest you either provide close-up imagery of at least some of the presented datasets, or better even provide video material for each dataset presented in your manuscript. Video material would allow a much better examination of the described phenomena and a much better understanding of them by readers. If you are not able to provide either video material or close-up imagery, please elaborate on why that is the case, as most high quality X-ray imaging studies in powder bed fusion published in recent years tend to do so for the mentioned reasons.

L.162: The causal link between bubble explosions and spatter ejection is not evident from the imagery in Fig. 6. The evaporation which induces the visible vapor depression is itself strong enough to cause the visible spatter ejection and provides an easier and alternative explanation. In that regard, plenty of X-ray studies exist that confirm the link between evaporation and spatter ejection (both in PBF-LB & EB). As already mentioned, please provide better imagery or video material such that the described behavior is easier to observe.

Fig. 6: The particle sizes visible in Fig. 6 do not match the powder particle size distribution of 15 to 45 μm stated in your methods section. In your radiographs, the particles seem to be roughly an order of magnitude larger than the used Al6061 powder. This indicates that the powder particles were actually molten during “sintering” which caused them to coalesce into larger particles. Please provide a detailed explanation on how the samples were prepared, including how the pre-sintered powder layers were created and applied to the thin-walled bulk samples used in your study.

Discussion

L192-194: The explanation of bubble formation is highly flawed from a physical perspective. First, you essentially attribute the formation of a bubble to the fact that the irradiated material liquefies and evaporates below the surface first which results in an accumulation of vapor pressure that leads to the formation of a bubble. While it might be true that the energy deposition is highest a few μm below the irradiated surface, the outgoing heat flux (as a result of thermal diffusion) is also much higher there as the heated volume is surrounded by highly conductive material from all sides. In contrast, the surface of the irradiated material can only conduct heat downwards. This is why liquefaction of material irradiated by an electron beam should still be expected to start from its surface, unless specific energy transport calculations suggest otherwise. Second, your assumption that the material reaches its boiling point first where energy deposition rate is highest is incorrect. The boiling point of a material strongly depends on the surrounding pressure. This means that even if the material gets heated from the inside, it does not necessarily start to boil there as well (or rather evaporate) as it is still surrounded by solid or liquid material. Since the surface of the irradiated material is in contact with the vacuum environment of your process chamber, it requires significantly lower temperatures to evaporate (or rather has a much lower boiling point) and thus should be expected to do so first. This means that while your explanation for the observed bubble formation behavior is not necessarily incorrect, it is not convincing unless necessary calculations or simulations are performed that show that evaporation is highly likely to initiate inside the irradiated material which would likely be a totally unprecedented discovery.

L.207-209 (+ L.214-217): You seem to essentially attribute all observed melt pool dynamics to the formation and explosion of bubbles with no consideration for other sources of hydrodynamic effects. Aside from the higher energy absorption efficiency and penetration depth provided by the electron beam, your experimental setup very closely resembles that of research groups from APS (Zhao et al. 2017, <https://doi.org/10.1038/s41598-017-03761-2>), SSRL (Calta et al. 2018, <https://doi.org/10.1063/1.5017236>) and DLS (Leung et al. 2018, <https://doi.org/10.1016/j.addma.2018.08.025>). These and some other groups have published extensive studies of melt pool dynamics in PBF-LB and should not be disregarded entirely on the basis of a difference in used heat source alone. Additionally, studies about melt pool dynamics in EBM from other research groups also exist and should be considered for discussion as well. The fact that your submitted article is entirely devoid of a comparison between your findings and the existing body of work in this field also seems questionable in

general. I therefore highly recommend a discussion of your work that considers findings by other research groups, especially considering the similarities in observed melt pool dynamics (aside from the presented bubble formation and explosion mechanism). This should also result in a much more nuanced understanding of the inducing effects of bubble formation and explosion discovered by your experiments.

L.237- 239: Is the vapor plume inside the depression actually strong enough to sufficiently scatter the electron beam to induce collapse? Be reminded that the high acceleration voltage is sufficient to allow the electron beam to penetrate fairly deep into solid material. How about performing a rough calculation of the penetration depth of the electron beam for reasonable vapor pressures, based on the same equations you used to calculate the energy deposition profile in Fig. 7a (by assuming that the vapor has the same composition as the solid, just at a much lower density)? This would contextualize your conclusion by some actual numbers (even if they are based on estimates). Additionally, your explanation that electron scattering causes keyhole collapse does not match the general observation that keyholes start to collapse from the surface through the formation of a constriction. How about considering material transport phenomena inside the melt as well?

L.285-287: The backwards melt flow visible in Fig. 9 is a fairly common phenomenon that does not rely on bubble explosion to occur. The backwards material ejection simply follows from the molten material having to flow around the depression, during which it gains momentum in the opposite direction of the moving electron beam/depression. Like previously stated, don't just attribute all observations to bubble formation and explosion without considering more commonly known and often likely alternatives.

L.293: Humping is a well known phenomenon in PBF-LB, where based on your provided explanation, bubble formation is not possible to occur. For reference, see Bhatt et al. 2023 (<https://doi.org/10.1016/j.addma.2023.103809>) for example.

L.308: Your statement that the bubble formation mechanism is not material-specific is incorrect, especially considering that there is no evidence for it. While electron beams can penetrate all metallic materials, the extent of their ability to do so differs drastically between alloys. Al6061 is arguably the material exhibiting the most amount of electron beam penetration (of all commonly processed EBM alloys) due to its low density (and low average atomic number). More commonly processed materials in EBM, like Cu, steels, Ni-base superalloys and refractory alloys offer much lower beam penetration depths and have (if your provided explanation for bubble formation is assumed to be correct) therefore a much lower chance of exhibiting bubble formation. Even a previously published article by the same group (Yuan et al. 2024, <https://doi.org/10.1016/j.addlet.2024.100239>) where Ti6Al4V and Al6061 was used to perform in-situ X-ray imaging experiments, shows no signs of bubble formation and explosion. This suggests that the observed phenomenon is, contrary to stated, highly material specific and likely unique to the combination of Al6061 and EBM and that it does not occur all the time, as Al6061 was melted in this study using the same sample geometry, beam power and scan velocity without exhibiting the bubble formation phenomenon (other melt pool dynamics seem to be similar to those reported here though).

L.314-316: The high beam penetration depth of EBM is arguably its biggest advantage over other additive manufacturing techniques, as it leads to a much better distribution of heat and lower evaporation during melting. Suggesting to remove this benefit to avoid bubble formation is highly counter-intuitive and would defy the purpose of EBM almost entirely.

Methods

L.333: Was any X-ray transparent confinement used, between the Al6061 samples were sandwiched? If not, did the melt pool stay confined between solid Al6061 during melting (i.e. did you make sure to create a melt pool thinner than the 1.2 mm sample thickness). Could you include an image of an Al6061 sample post experimental procedure in Fig. 1, such that the reader gets an expression of how melting affects the sample?

L.342-343: As previously stated, could you please provide a spot size definition for your electron beam diameter of 200 μm ?

References

L.448-449: Reference 34 (Bidola, P. M. et al. In situ synchrotron...) is wrongly referenced. The first author of this publication should be "Semjatov, N." not "Bidola, P. M."

Final remark

Assuming the described methodology is complete, the in-situ synchrotron imaging study was performed at room temperature. In EBM, processes are always performed at an elevated build temperature to enable sintering of powder particles with minimal energy input, in order to minimize the risk of smoke and reduce accumulation of thermal stress. As a result, solidification times are usually a few orders of magnitude larger than in PBF-LB which is one of the reasons why melt pool dynamics can differ drastically between the two techniques. Since elevated temperatures were not used in your study, solidification behavior should actually be more similar to that encountered in PBF-LB than in EBM (PBF-EB). How do you think does this affect the transferability of your findings to industrial PBF-EB processes? Would you expect to see the same melt pool dynamics, bubble formation and explosion behavior, as well as humping formation or would you expect differences?

Aside from my remarks, I find the presented experimental findings highly interesting and am looking forward to their publication once the necessary adjustments have been made.

Sincerely,

Nick Semjatov

Version 1:

Reviewer comments:

Reviewer #1

(Remarks to the Author)

The authors responded well and made revisions to the reviewer's comments. The newly revised paper focuses on Al6061 aluminum alloys, which is more rigorous and comprehensive, and also provides reference and inspiration for EBM processes of other materials. Therefore, I believe that the paper can be accepted.

Reviewer #2

(Remarks to the Author)

The authors have addressed the reviewers' comments in an appropriate and satisfactory way. In particular, the hedging of the interpretations of the observations to the actual experiment improve the manuscript. Further the clarification of the phenomenological nature, as well as the identification of the limitations and scope of the work benefit the manuscript.

The presented results and their interpretation are more clear and categorized. The limitations of the work and the directions of future work are pointed out, which strengthens the contribution of the work to PBF-EB research.

The authors should include some markers/guidance to the eye to ensure that there is no confusion about the observations in the videos.

With including this adjustment, the manuscript gives a clear and coherent presentation of phenomenological observations and situates these in the research field. It is therefore suitable for consideration for publication.

Reviewer #3

(Remarks to the Author)

This revision leaves the manuscript in a much better state. All of my remarks have been sufficiently addressed. With the inclusion of video material, the authors provide much stronger and easier to follow evidence for their reported observations (the bubble formation and explosion phenomenon is much easier to see this way). The description of experimental procedures is also significantly improved and the description of scientific impact and relevance to the field of PBF-EB appropriately adjusted. There is no further revision needed from my point of view. I am looking forward to the publication of this study.

Sincerely,
Nick Semjatov

Response to Reviewer's comments

We deeply appreciate the editor's and reviewers' time and effort in providing constructive comments and valuable advice on our manuscript. Over the past month, we have made our best efforts to address all the insightful comments and improve the paper accordingly. Below, we provide a detailed description of the corrective actions in the revised manuscript and our point-by-point responses to each reviewer's comments (the revised sections are highlighted in different colors).

Reviewer #1 (Remarks to the Author):

This manuscript used in-situ high-speed high-resolution synchrotron X-ray imaging technology to observe the formation and explosion of subsurface bubbles in the molten pool during the electron beam powder bed fusion (EB-PBF) process of aluminum alloy Al6061 and investigated the mechanism of the subsurface bubble formation and the effects on the molten pool instabilities, and the morphology of the melt track. Since it is the first time that subsurface bubbles have been directly observed during the melting and deposition process of EB-PBF, this manuscript is of great significance for the research of EB-PBF technology. Although the classical EB process theory suggested that the energy absorption peak of the EB is several to tens of micrometers below the surface of the material, there is no experimental observation directly indicating the formation of micro-bubbles and its explosions below the surface. Moreover, for the aluminum alloy used in this paper as the experimental material, the frequency of bubble formation and explosion is very high, almost existing under all energy density conditions. Therefore, the findings of this paper are original.

Response 1-1: We sincerely thank the reviewer for the thorough evaluation of our manuscript and the encouraging comments. All revisions made in response to the reviewer's comments have been highlighted in blue in the revised manuscript.

This manuscript is valuable and an important reference for understanding the EB-PBF process, but more evidence is needed to prove that its conclusions are universal and can guide the process development for other materials. The main reason is that the aluminum alloy used in this experiment is an uncommon material for EB-PBF process with low density, low melting point and boiling point, which makes the subsurface bubble easy to form. On the contrary, the common

materials for EB-PBF, such as titanium alloys, titanium aluminum alloys and superalloys, have higher melting points and boiling points, and higher difficulty to form the bubbles. So, more evidence is required to prove the existence of bubble formation and explosion phenomena in these materials as well. Here, the factors need to be considered not only the higher melting and boiling points of the material, but also the higher material density, which makes the position of the EB energy absorption peak closer to the surface. Therefore, more experimental research may be needed to verify whether the bubble formation and explosions in EB-PBF are “not material-specific and is expected to manifest across all material systems” (Page 15, line 308). Before the author provides more evidence, it is best for the title and conclusion of this manuscript to focus on the EB-PBF process for aluminum alloys.

Response 1-2: We thank the reviewer for the encouraging and insightful comments. We agree that bubble formation and explosion behaviors may be material-specific, as variations in physical properties such as melting point, boiling point, atomic number, and density can significantly influence energy absorption and vaporization behavior during EB-PBF. Further experimental investigations on other alloy systems (e.g., Ti- and Ni-based alloys) are indeed necessary to verify the generality of this phenomenon.

Accordingly, we have revised the *Title, Abstract, Introduction, Discussion, and Conclusion* to focus specifically on aluminum alloys, while clearly acknowledging the need for further validation across additional material systems. The revised content is as follows.

Title (Lines 1 and 2): Bubble explosion induced melt pool instabilities in electron beam melting of aluminum alloy

Abstract (Lines 18-20): Here, using in-situ high-speed synchrotron X-ray imaging, we reveal that bubble explosions in Al6061 during EBM induce melt pool instabilities contributing to defect formation.

Introduction (Lines 58-60): In this study, taking advantage of this in-situ characterization system, we discovered that bubble explosions during EBM in Al6061, unique to EBM, destabilize the melt pool and contribute to multiple types of defects.

Discussion: We deleted the previous statement: “*The melt pool instabilities induced by bubble explosion originate from the energy deposition characteristics inherent to the high-energy electron*

beam. This mechanism is not material-specific and is expected to manifest across all material systems. However, the energy absorption behavior, including the electron beam peak energy deposition depth and penetration depth, is highly material dependent.”

Conclusion (Lines 343-347): This work is expected to inspire the electron beam additive manufacturing and welding community to advance modeling efforts for a quantitative understanding of bubble formation, explosion, and their effects, and to experimentally investigate, through in-situ monitoring, the occurrence of subsurface bubble explosions in high-atomic-number materials, thereby deepening the overall understanding of electron beam melting.

Another point worth discussing is, the manuscript mentioned in Page 16 line 314, the strategy of an appropriate EB accelerating voltage to shift the peak energy deposition closer to the surface to suppress melt pool instabilities induced by bubble explosion. Actually, shifting the energy deposition closer to the surface usually needs to lower the acceleration voltage of the EB. But existing research (Ref. 1) has shown that although increasing the EB acceleration voltage will deepen the position of energy absorption, it can also increase the total depth of EB energy absorption effectively, and make energy absorption more uniform at different depths. Meanwhile, keeping the constant power, the higher acceleration voltage will make the intensity of the EB low, which can further reduce the energy density of the EB energy absorption peak, and should be beneficial for reducing the bubble formation as well. Additionally, the exiting research (Ref. 1) indicated already that higher acceleration voltage of the EB could obtain higher density and fewer defects, which may be partially due to the reduction of bubble formation.

Response 1-3: We thank the reviewer for this insightful comment regarding the influence of electron beam accelerating voltage. We agree that increasing the accelerating voltage could deepen and broaden the region of energy absorption, resulting in more uniform energy distribution. Additionally, at constant beam power, a higher accelerating voltage reduces beam intensity and peak energy density, which could suppress bubble formation and improve part density, as reported in the literature.

In the present work, the primary focus is on revealing the existence of subsurface bubble formation and explosion during the electron beam melting of Al6061. A systematic investigation of bubble formation and explosion under different accelerating voltages, and their effects on melt pool dynamics, will be conducted in future studies. In the revised version, we have removed the

discussion related to accelerating voltage strategy (Page 16, line 314 in the original manuscript) and added the following updated discussion.

Lines 333-336: Furthermore, the effect of electron beam acceleration voltage on bubble explosion behavior and melt pool dynamics warrants systematic investigation, as higher acceleration voltages deepen and homogenize energy deposition, potentially reducing the propensity for subsurface bubble explosion⁵⁸.

58. Li, H., Yu, Y., Li, Y. & Lin, F. Effects of the higher accelerating voltage on electron beam powder-bed based additive manufacturing of Ti6Al4V alloy. *Addit. Manuf.* 50, 102579 (2022).

In addition to the above issues, the experimental methods, data analysis, interpretation and conclusions of the manuscript are quite appropriate.

Response 1-4: We thank the reviewer for the positive feedback.

Finally, one more suggestion that the authors may be interested in. Based on the engineering practice, the formation and explosion of the subsurface bubbles may not only cause melt pool instabilities, lack-of-fusion defects, and melt track irregularity, but also may trigger the phenomenon of "smoking", which wasn't mentioned in the manuscript yet. Existing research (Ref. 2) has revealed that when EB encounters powder particles above the powder bed, it will charge the particles and be dispersed onto the powder bed outside the focal spot, and the charged particles will also fall down onto the powder bed, thereby triggering smoking. The spatters, generated by the subsurface bubble explosions, may be a trigger to cause smoking as well if they can fly to a higher altitude and coincidentally encounter the EB. This may explain the sudden occurrence of smoking during selective melting with a focused EB in the EB-PBF process. If recognized by the authors, it can be added to the background introduction of the manuscript.

Response 1-5: We thank the reviewer for this thoughtful and insightful suggestion. We agree that explosion-induced spattering could potentially contribute to the smoking phenomenon in EB-PBF. As described in the mentioned studies, smoking can arise when charged particles generated above the powder bed are dispersed by the electron beam and subsequently fall back, disturbing the powder layer. Spatters produced by subsurface bubble explosions may similarly interact with the electron beam, become charged, and thus act as triggers for smoking events.

We have added a discussion of this possibility in *Discussion* section of the revised manuscript as a potential link between bubble explosion–induced spattering and the onset of smoking.

Lines 226-230: Previous research has shown that when the electron beam interacts with powder particles above the powder bed, these particles can become charged and dispersed beyond the focal spot before eventually falling back onto the powder bed, thereby triggering smoking⁵⁴. Similarly, spatters produced by subsurface bubble explosions may interact with the electron beam, become charged, and consequently act as potential initiators of smoking events.

54. Wang, D., Zhao, D., Liang, X., Li, X. & Lin, F. Multiple stages of smoking phenomenon in electron beam powder bed fusion process. *Addit. Manuf.* 66, 103434 (2023).

Reference:

1. "Effects of the higher accelerating voltage on electron beam powder-bed based additive manufacturing of Ti6Al4V alloy." *Additive Manufacturing* 50 (2022): 102579.
2. "Multiple stages of smoking phenomenon in electron beam powder bed fusion process." *Additive Manufacturing* 66 (2023): 103434.

Response 1-6: We thank the reviewer for providing these valuable references. They are highly relevant and have strengthened both the current discussion and the context for future work. We have incorporated the suggested citations into the revised manuscript.

Reviewer #2 (Remarks to the Author):

The authors report the electron beam additive manufacturing specific process instability caused by bubble explosion. Further, the observations are connected and explained by the electron beam matter interaction.

The reported results provide important insights that will facilitate improved process control and final quality of manufactured components.

Especially figure 7 and 8 provide sound evidence of the claims.

Response 2-1: We are grateful to the reviewer for recognizing the value and merit of our study. All revisions made in response to your comments have been highlighted in green in the revised manuscript.

1. Further details on the prepared samples can be provided.

How was the sintering performed? Could there be gas or humidity entrapped in the samples?

Response 2-2: We thank the reviewer for this helpful suggestion. Additional details regarding sample and sintered-layer preparation have been included in the revised manuscript (lines 351–364) and are summarized below.

Pre-sintered powder-bed Al6061 samples were prepared by sintering one layer of Al6061 powders (15–45 μm) on a commercial Al6061 plate using laser heating under an argon atmosphere to minimize oxidation. After pre-sintering, the samples were cut, and the exposed side surfaces were ground to 800 grit and then cleaned in an ultrasonic bath with ethanol to remove potential surface contaminants. For the bare plate samples, after cutting, the side and top surfaces of samples were similarly ground to 800 grit and then cleaned in an ultrasonic bath with ethanol to remove potential surface contaminants. During the electron beam melting experiments, the samples were placed inside a vacuum chamber at a pressure of 10^{-5} Torr, allowing any residual surface gas to be released.

During X-ray imaging of electron beam melting process, no surface instabilities were observed. All bubbles observed and reported in this study were generated beneath the liquid metal surface, such as in Fig. 8, and were nucleated within the front keyhole wall rather than on the top surface in any of our data, demonstrating that the samples did not entrap gas or humidity.

To avoid confusion for readers, clarification on sample and sintered-layer preparation was added to the revised manuscript, as shown below:

Lines 351-364: Al6061 was selected as the material for this study because of its good X-ray transparency for in-situ high-speed X-ray imaging experiments⁵². Two types of samples were prepared for the experiments: bare substrates and pre-sintered powder-bed samples. The bare substrate samples were sectioned from commercial Al6061 plates (purchased from McMaster). Since powder bed sintering is not the focus of this study, to accelerate the beamline experiments,

the powder layer was pre-sintering before loading into the chamber. The pre-sintered powder-bed Al6061 samples were prepared by sintering one layer of Al6061 powders (15–45 μm) on a commercial Al6061 plate using laser heating under an argon atmosphere to minimize oxidation. As observed in the X-ray images, the powder particles appear to have partially melted and coalesced into larger agglomerates, with apparent diameters of approximately 200 μm . This coalescence results from localized laser melting during the sintering process rather than the use of larger initial powder particles. Both types of Al6061 samples were cut into pieces with dimensions of 58 mm (length) x 10 mm (width) x 1.2 mm (thickness in the X-ray penetration direction), with the powdered surface facing the incident beam during experiments.

2. In line 42-43 the authors claim that boiling is always present in ebm processing. But steering the processing parameters allows for selection of keyhole or conduction mode melt pools. It is not necessary to process with keyhole.

Response 2-3: We agree with the reviewer that electron beam powder bed fusion is typically conducted in the conduction mode, often without the formation of a vapor depression zone. The word “boiling” has been removed.

Lines 42 and 43: In EBM, accelerated electrons bombard the pre-sintered powder bed, converting their kinetic energy into heat and causing localized melting.

3. The sentence 249-250 should be rephrased.

Response 2-4: We thank the reviewer for pointing out this issue. The sentence has been rephrased:

Lines 270 and 271: On the inclined front wall of the vapor depression, instabilities induced by bubble explosion lead to the formation of protrusions (Fig. 8b).

4. Line 308-309 is it also expected in tungsten?

Response 2-5: We thank the reviewer for this valuable comment. Based on our current results, we cannot determine whether bubble formation and explosion can occur in tungsten. The bubble formation and explosion behavior is likely to be material-specific, and it remains unknown whether it occurs in high-melting point, high-density materials such as tungsten. Significant differences in physical properties, including melting and boiling points, vapor pressure, atomic number, and

density, can substantially influence the energy absorption profile and vaporization dynamics in EB-PBF.

To clarify this point, we have revised the *Title, Abstract, Introduction, Discussion* and *Conclusion* to explicitly focus on aluminum alloys and to indicate that further experimental studies on other alloy systems (e.g., W-based materials) are required to evaluate the generality of the subsurface bubble formation phenomena.

Title (Lines 1 and 2): Bubble explosion induced melt pool instabilities in electron beam melting of aluminum alloy

Abstract (Lines 18-20): Here, using in-situ high-speed synchrotron X-ray imaging, we reveal that bubble explosions in Al6061 during EBM induce melt pool instabilities contributing to defect formation.

Introduction (Lines 58-60): In this study, taking advantage of this in-situ characterization system, we discovered that bubble explosions during EBM in Al6061, unique to EBM, destabilize the melt pool and contribute to multiple types of defects.

Discussion: We deleted the previous statement: “*The melt pool instabilities induced by bubble explosion originate from the energy deposition characteristics inherent to the high-energy electron beam. This mechanism is not material-specific and is expected to manifest across all material systems. However, the energy absorption behavior, including the electron beam peak energy deposition depth and penetration depth, is highly material dependent.*”

Conclusion (Lines 343-347): This work is expected to inspire the electron beam additive manufacturing and welding community to advance modeling efforts for a quantitative understanding of bubble formation, explosion, and their effects, and to experimentally investigate, through in-situ monitoring, the occurrence of subsurface bubble explosions in high-atomic-number materials, thereby deepening the overall understanding of electron beam melting.

Reviewer #3 (Remarks to the Author):

The authors present a highly novel and compelling study about a newly discovered bubble formation and explosion phenomenon that occurs during melt pool formation within Al6061 samples using EBM (better referred to as PBF-EB nowadays), enabled by high-speed synchrotron imaging with a specialized in-situ sample environment. The discovered phenomenon is novel and has never been reported on, making it highly suitable for publication.

Response 3-1: Thank you very much for the thorough evaluation of our manuscript. We sincerely appreciate your evaluation and encouraging comments. The changes in the manuscript relevant to your comments have been highlighted in purple.

Unfortunately, the presentation of the phenomenon, as well as its discussion in its current state, is highly flawed and requires substantial revision. First of all, the lack of supplementary video material (which is often included with such in-situ studies) as well as appropriately processed imagery focusing on the phenomenon itself, make it highly difficult for the reader to confirm the claims and descriptions in the written paragraphs. The imagery should be improved for better clarity, focusing on the bubble explosion phenomenon and ideally video material should also be provided if possible.

Response 3-2: We thank the reviewer for this helpful suggestion. Ten X-ray videos of the electron beam melting process have been included in the Supplementary Materials, with each melting sequence presented in the manuscript paired with its corresponding video. The quality and design of the images have been improved.

Next, the provided explanations for observed phenomena are highly superficial and lack sufficient theoretical and mathematical considerations to be convincing. Here, the authors seem to provide explanations that best suit the narrative of their article instead of considering the established complexity and nuance of melt pool dynamics in powder bed fusion additive manufacturing. Additionally, the authors attribute essentially all observed melt pool dynamics (keyhole shape, periodic collapse, melt pool backflow, spattering, etc.) to the bubble explosion phenomenon, despite there being an established body of scientific work (predominantly in-situ imaging of laser powder bed fusion, as well as CFD simulations) providing much more elaborate and convincing explanations for many of the described phenomena. The article, in its current state,

is also entirely devoid of a discussion of observed phenomena that considers previously established descriptions of melt pool dynamics in powder bed fusion AM which seems highly inappropriate, as many of the observed phenomena very closely resemble the findings of other research groups and studies. Since the provided explanations for observed phenomena are mostly phrased as factual truths and not as working hypotheses, the authors are advised to provide a much more elaborate analysis of their experimental findings to support their claims, in order to demonstrate a consideration of established research in this field, especially considering the novelty of their discovered phenomenon. Alternatively, if this turns out to be outside the scope of this article, it should be made clear that the bubble explosion phenomenon could potentially be a contributing factor in most of the observed melt pool dynamics and not their sole cause.

Response 3-3: We sincerely thank the reviewer for this valuable and constructive comment.

Regarding the first point:

In this study, our primary goal is to reveal a previously unreported phenomenon that may occur during electron beam melting and to demonstrate its influence on melt pool dynamics. We agree with the reviewer that more quantitative modeling and calculation are required to further elucidate the underlying mechanisms of subsurface bubble formation, explosion, and their influence. Our objective for this paper is to bring this phenomenon to the attention of researchers and engineers in the field, so that future efforts can collectively advance understanding through improved equipment design, process optimization, alloy development, and numerical modeling. Accordingly, we have clarified this point in the revised manuscript:

Conclusion (Lines 343-347): This work is expected to inspire the electron beam additive manufacturing and welding community to advance modeling efforts for a quantitative understanding of bubble formation, explosion, and their effects, and to experimentally investigate, through in-situ monitoring, the occurrence of subsurface bubble explosions in high-atomic-number materials, thereby deepening the overall understanding of electron beam melting.

Regarding the second point:

We agree that the observed melt pool dynamics cannot be solely attributed to bubble formation and explosion. Other hydrodynamic effects—including Marangoni convection, recoil pressure,

surface tension gradients, and thermocapillary-driven flow—also play essential roles in melt pool dynamics.

In the updated *Results and Discussion* sections, we provide a more comprehensive analysis that situates our findings within the context of previous studies on melt pool dynamics under both laser and electron beam conditions. Simultaneously, in addition to bubble explosion, we now consider others factors that can influence the melt pool dynamics, including vapor pressure, evaporation, Rayleigh–Taylor instability and related mechanisms. We have added the following discussion to the revised manuscript:

Regarding the evolution of the melt pool and keyhole:

Lines 112-116: At this stage, the rapid expansion of the melt pool and keyhole is primarily driven by rising vapor pressure within the depression, which displaces molten material and enhances heat flux into the surrounding solid, similar to the mechanism of keyhole drilling observed in laser powder bed fusion⁴⁶. Bubble explosions may further amplify local pressure fluctuations, promoting additional keyhole deepening.

46. Cunningham, R. et al. Keyhole threshold and morphology in laser melting revealed by ultrahigh-speed x-ray imaging. *Science* 363, 849–852 (2019).

Lines 234-235: These dynamics contribute to reshaping the melt pool and vapor depression through two coupled mechanisms.

Regarding spattering:

Lines 165-177: The dynamics of spattering under the influence of subsurface bubble formation and explosion were also investigated. Since spatter primarily arises from interactions between metallic vapor and the melt pool surface or powder bed, pre-sintered powder bed samples were used to study interactions among metal vapor, melt pool and powder. Following subsurface bubble explosions (Fig. 6a and b), the melt pool became unstable within the subsequent frames, ejecting a liquid ligament from the explosion site followed by spatters (Fig. 6c-f and Supplementary Movie 7). To facilitate clearer identification of bubble explosions and the associated spatters, the original images were processed and are shown in Fig. 6g and h. An additional example of bubble-explosion–induced spattering with preheating prior to melting is provided in Supplementary Note

2, Supplementary Fig. 2 and Supplementary Movie 8. It is noted that strong evaporation at the vapor depression without bubble explosion can also drive spatter ejection, as supported by numerous X-ray studies in both laser and electron beam powder bed fusion processes^{34,47}. The bubble explosions observed here represent an additional mechanism that can contribute to spatter formation.

34. Semjatov, N., Wahlmann, B. & Carolin, K. Multiple interaction electron beam powder bed fusion for controlling melt pool dynamics and improving surface quality. *Addit. Manuf.* 90, 104316 (2024).

47. Guo, Q. et al. Transient dynamics of powder spattering in laser powder bed fusion additive manufacturing process revealed by in-situ high-speed high-energy x-ray imaging. *Acta Mater.* 151, 169–180 (2018).

Regarding melt flow:

Line 300-305: Second, the backward melt flow observed in Fig. 9c, a phenomenon commonly reported in laser powder bed fusion, typically arises as molten metal circulates around the vapor depression and gains momentum opposite to the scanning direction⁵⁷. Bubble explosions occurring at the front wall may introduce additional momentum to the backward flow by generating transient vapor jets and surface waves that propagate along the vapor–liquid interface.

57. Guo, Q. et al. In-situ full-field mapping of melt flow dynamics in laser metal additive manufacturing. *Addit. Manuf.* 31, 100939 (2020).

Lines 306-311: As the melt flow propagates backward, periodic undulations develop along the melt pool surface due to the Rayleigh–Taylor instability, as previously reported in laser and electron beam powder bed fusion^{34,41}. As the solidification front advances, these undulations evolve into humping structures along the melt track. In electron beam melting, intermittent momentum impulses generated by bubble explosions further amplify this instability by perturbing the melt pool surface. The combined effects of Rayleigh–Taylor and explosion-induced instabilities thus promote periodic undulations, ultimately leading to the formation of humping structures along the melt track (Fig. 9d and Supplementary Movie 5).

34. Semjatov, N., Wahlmann, B. & Carolin, K. Multiple interaction electron beam powder bed fusion for controlling melt pool dynamics and improving surface quality. *Addit. Manuf.* **90**, 104316 (2024).

41. Bhatt, A. et al. In situ characterisation of surface roughness and its amplification during multilayer single-track laser powder bed fusion additive manufacturing. *Addit. Manuf.* **77**, 103809 (2023).

Considering the novelty of the presented findings, the article is definitely worthy of publication once these general concerns, as well as those described below have been addressed.

Response 3-4: We thank the reviewer for offering positive feedback and listing the issues with the manuscript.

Abstract

1.19: The statement that bubble explosions are intrinsic to EBM is overselling the findings, as a single study with a single material is not sufficient evidence for the introduction of bubble explosions as an intrinsic mechanism of this AM technology.

Response 3-5: We thank the reviewer's critical comment. We agree with the reviewer. The word "intrinsic" has been removed.

Lines 18-20: Here, using in-situ high-speed synchrotron X-ray imaging, we reveal that bubble explosions in Al6061 during EBM induce melt pool instabilities contributing to defect formation.

Introduction

1.44: EBM can create vapor depressions under certain conditions but does not necessarily lead to their formation in general.

Response 3-6: We agree with the reviewer. The words "vapor depression" have been removed.

Line 42: This interaction generates a melt pool.

L.60: Similar to the Abstract, “intrinsic and unique to EBM” is overstating the significance of the presented findings.

Response 3-7: The word “intrinsic” has been removed. The revised text reads:

Lines 58-61: In this study, taking advantage of this in-situ characterization system, we discovered that bubble explosions during EBM in Al6061, unique to EBM, destabilize the melt pool and contribute to multiple types of defects.

Results

L.78: You do not provide a spot size definition for their stated electron beam spot size. Is the spot size 200 μm FWHM (Full Width Half Max), FWHP (Full Width Half Power), $D4\sigma$ (4 standard deviations) or some other spot size definition?

Response 3-8: We thank the reviewer for this insightful comment. In the experiments, the electron beam has a diameter of approximately 200 μm at the working plane, defined as the full width at half maximum (FWHM) of a Gaussian energy distribution. The beam information was measured using a slit-scan method, in which the beam was stepped across a slit of known width while measuring the transmitted current to determine the rise/fall regions. Then the spot size was calculated using the FWHM metric. This clarification has been incorporated into the revised manuscript to ensure accuracy and consistency in the beam specification.

Lines 80-82: The experiments were conducted in a vacuum chamber using a beam with a diameter of approximately 200 μm (full width at half maximum (FWHM) of a Gaussian energy distribution).

Lines 371-375: In the experiments, the electron beam has a diameter of approximately 200 μm at the working plane, defined as the full width at half maximum (FWHM) of a Gaussian energy distribution. The beam information was measured using a slit-scan method, in which the beam was stepped across a slit of known width while measuring the transmitted current to determine the rise/fall regions. Then the spot size was calculated using FWHM metric.

L.101: You state that the rapid melt pool and keyhole expansion are the result of bubble explosions, yet there is no reason for this being the case. The increase in both can easily be

attributed to increasing vapor pressures which result in a displacement of the molten material and lead to an increase in heat flux into the still solid substrate.

Response 3-9: We appreciate the reviewer for this valuable comment and agree that the rapid expansion of the melt pool and keyhole can indeed be attributed to increasing vapor pressure, which displaces molten material and enhances heat transfer into the surrounding solid. We have revised the relevant text to clarify that bubble explosion is not the sole cause but rather a possible contributing factor. The content has been revised manuscript:

Lines 106-107: At this stage, violent bubble explosions can contribute to the rapid deepening and widening of the keyhole.

Lines 112-116: At this stage, the rapid expansion of the melt pool and keyhole is primarily driven by rising vapor pressure within the depression, which displaces molten material and enhances heat flux into the surrounding solid, similar to the mechanism of keyhole drilling observed in laser powder bed fusion⁴⁶. Bubble explosions may further amplify local pressure fluctuations, promoting additional keyhole deepening.

46. Cunningham, R. et al. Keyhole threshold and morphology in laser melting revealed by ultrahigh-speed x-ray imaging. *Science* 363, 849–852 (2019).

L. 107-110: Again, there is no necessary causal link between the shown vapor depression formation and the effect of bubble formation and explosion. The widening of the depression at the beginning could simply be the result of material transport phenomena. The fact that keyholes get more narrow as they deepen (e.g. common in laser powder bed fusion) can be explained by the necessarily higher vapor pressures to maintain them. Since the electron beam most likely follows a gaussian distribution, evaporation is strongest in its center and thus the keyhole gets more narrow as it deepens as a consequence, as only the energy flux density in the beam's center is sufficient to displace the melt. While the bubble explosions could be a contributing factor, there is no compelling evidence to suggest that it dominates the melt pool dynamics. How many datasets did show the behavior presented in Fig. 3f – h, exactly? Is the shown example statistically significant? Have any calculations been performed to estimate the force exerted by bubble explosions onto the surrounding melt, compared to the force exerted by evaporation directly? Is there an underlying

physical model that strongly suggests that bubble explosion could dominate melt pool displacement at all? If so, why was it not included in the manuscript?

Response 3-10: We thank the reviewer for this insightful comment and agree that vapor depression formation and evolution can primarily result from material transport phenomena the increasing vapor pressure within the depression. We have revised the relevant discussion to clarify that bubble explosions are not the sole cause of the deepening and widening of the keyhole, but rather an additional contributing factor.

In the revised manuscript, we now state:

Lines 112-116: At this stage, the rapid expansion of the melt pool and keyhole is primarily driven by rising vapor pressure within the depression, which displaces molten material and enhances heat flux into the surrounding solid, similar to the mechanism of keyhole drilling observed in laser powder bed fusion⁴⁶. Bubble explosions may further amplify local pressure fluctuations, promoting additional keyhole deepening.

46. Cunningham, R. et al. Keyhole threshold and morphology in laser melting revealed by ultrahigh-speed x-ray imaging. *Science* 363, 849–852 (2019).

Regarding the representativeness of the data, we thank the reviewer for the insightful comment and agree that reproducibility and statistical relevance are essential. The experiments were performed using our custom-built electron beam melting system at the Advanced Photon Source, Argonne National Laboratory, across four independent beamtime sessions (February 1–7, 2022; June 1–9, 2022; October 25–28, 2022; and March 8–13, 2023), totaling 624 experimental hours and producing 318 datasets. The phenomena of bubble formation and explosion were consistently observed in both Al6061 and AlSi10Mg alloys. Among the 41 stationary melting datasets, 15 were performed under beam energy conditions sufficient to generate a keyhole. The bubble explosion and keyhole evolution behaviors were reproducibly captured across all these 15 stationary melting experiments conducted under comparable conditions.

Regarding the force estimation and the physical model, we acknowledge that a quantitative analysis of the pressure dynamics induced by bubble explosions and a detailed physical model of the associated melt pool response are beyond the current scope of this work. We have removed the previous interpretation based solely on bubble formation and explosion and have expanded the

discussion to incorporate mechanisms that align more closely with observations from both EBM and LPBF systems. This limitation and the need for future modeling efforts have been clearly stated in the revised manuscript.

Lines 112-116: At this stage, the rapid expansion of the melt pool and keyhole is primarily driven by rising vapor pressure within the depression, which displaces molten material and enhances heat flux into the surrounding solid, similar to the mechanism of keyhole drilling observed in laser powder bed fusion⁴⁶. Bubble explosions may further amplify local pressure fluctuations, promoting additional keyhole deepening.

46. Cunningham, R. et al. Keyhole threshold and morphology in laser melting revealed by ultrahigh-speed x-ray imaging. *Science* 363, 849–852 (2019).

Conclusion (Lines 343-347): This work is expected to inspire the electron beam additive manufacturing and welding community to advance modeling efforts for a quantitative understanding of bubble formation, explosion, and their effects, and to experimentally investigate, through in-situ monitoring, the occurrence of subsurface bubble explosions in high-atomic-number materials, thereby deepening the overall understanding of electron beam melting.

L.110-113: Based on Fig. 3j, the onset of keyhole collapse seems to be attributable to the cessation of bubble explosions. Is this assumption correct? If so, be aware that keyhole collapse is a phenomenon not unique to EBM and does happen in PBF-LB as well where this bubble explosion behavior has never been observed. See the study of Qu et al. (<https://doi.org/10.1016/j.addlet.2022.100068>) for reference. The periodic collapse can be explained by Marangoni convection induced material transport in the keyhole surrounding melt pool, as well as fluctuations in electron beam irradiation due to changes in beam incidence angles and the effect of surface tension on the shape of the keyhole.

Response 3-11: We thank the reviewer for this insightful comment. We would like to clarify that the onset of keyhole collapse is not attributed to the cessation of bubble explosions. We agree with the reviewer that the periodic collapse of the keyhole results from the complex interplay of Marangoni convection, recoil pressure, and surface tension, which should be consistent with the mechanisms observed in both EBM and PBF-LB processes, as also discussed in the referenced study by Qu et al. (2022).

To avoid any possible misunderstanding or confusion for readers, we have removed the annotation in Fig. 3j that might have implied a direct causal relationship between bubble explosions and keyhole collapse, as shown below. In addition, the discussion about keyhole periodic collapse has been added.

Lines 256-261: During melting, recoil pressure from the dense metallic vapor drives the surrounding liquid outward, forming a deep and narrow cavity (Fig. 2i and j). As melting continues, the interplay among thermocapillary force, Marangoni convection, recoil pressure, and surface tension, similar to the mechanisms observed in laser powder bed fusion^{55,56}, induces keyhole instability. When the keyhole collapses, the high-energy electron beam reinitiates intense evaporation in the newly formed shallow keyhole, driving a cyclic collapse–regrowth instability (Fig. 2k–o).

55. Qu, M. et al. Mitigating keyhole pore formation by nanoparticles during laser powder bed fusion additive manufacturing. *Addit. Manuf. Lett.* 3, 100068 (2022).

56. Zhao, C. et al. Critical instability at moving keyhole tip generates porosity in laser melting. *Science* 370, 1080–1086 (2020).

Fig. 3. Initiation of keyhole and melt pool under the influence of subsurface explosion during electron beam stationary melting with an electron beam power of 426 W. (a-c) Stage 1: Initial formation of a melt pool during electron beam stationary melting. (d-j) Stage 2: Subsurface bubble formation and explosion. At this stage, violent bubble explosions can contribute to the rapid deepening and widening of the keyhole. (j-o) Stage 3: Periodic keyhole oscillation. In this stage, the keyhole depth starts to fluctuate periodically. p Melt pool and keyhole dimensions over time.

Fig.4 & 5: The bubble formation and explosion events are extremely difficult to discern from the provided radiography data, so much so that the reader is forced to believe the described dynamics without being able to confirm them through the provided imagery. I highly suggest you

either provide close-up imagery of at least some of the presented datasets, or better even provide video material for each dataset presented in your manuscript. Video material would allow a much better examination of the described phenomena and a much better understanding of them by readers. If you are not able to provide either video material or close-up imagery, please elaborate on why that is the case, as most high quality X-ray imaging studies in powder bed fusion published in recent years tend to do so for the mentioned reasons.

Response 3-12: We thank the reviewer's critical comment. Ten X-ray videos of electron beam melting have been included in the Supplementary Materials, with each melting sequence presented in the manuscript paired with its corresponding video.

L.162: The causal link between bubble explosions and spatter ejection is not evident from the imagery in Fig. 6. The evaporation which induces the visible vapor depression is itself strong enough to cause the visible spatter ejection and provides an easier and alternative explanation. In that regard, plenty of X-ray studies exist that confirm the link between evaporation and spatter ejection (both in PBF-LB & EB). As already mentioned, please provide better imagery or video material such that the described behavior is easier to observe.

Response 3-13: We thank the reviewer for this valuable and insightful comment. We agree that strong evaporation at the vapor depression can indeed drive spatter ejection, as supported by numerous X-ray studies in both laser and electron beam powder bed fusion processes. Our observations suggest that subsurface bubble explosions can be an additional mechanism that can contribute to spatter formation. The related description has been revised accordingly in the manuscript.

We have added the corresponding video (Supplementary Movie 7) and an additional example of bubble-explosion-induced spattering in the supplementary material. The new figure and its accompanying explanation have been added to the Supplementary Materials (Supplementary Note 2, Supplementary Fig. 2 and Supplementary Movie 8).

Lines 172-177: An additional example of bubble-explosion-induced spattering with preheating prior to melting is provided in Supplementary Note 2, Supplementary Fig. 2 and Supplementary Movie 8. It is noted that strong evaporation at the vapor depression without bubble explosion can also drive spatter ejection, as supported by numerous X-ray studies in both laser and electron beam

powder bed fusion processes^{34,47}. The bubble explosions observed here represent an additional mechanism that can contribute to spatter formation.

34. Semjatov, N., Wahlmann, B. & Carolin, K. Multiple interaction electron beam powder bed fusion for controlling melt pool dynamics and improving surface quality. *Addit. Manuf.* 90, 104316 (2024).

47. Guo, Q. et al. Transient dynamics of powder spattering in laser powder bed fusion additive manufacturing process revealed by in-situ high-speed high-energy x-ray imaging. *Acta Mater.* 151, 169–180 (2018).

Supplementary Note 2: Dynamics of spattering

In addition to the spattering behavior associated with bubble formation and explosion shown in Fig. 6 of the main text, a similar sequence is presented in Supplementary Fig. 2 and Supplementary Movie 8. Immediately after a subsurface bubble explosion (Supplementary Fig. 2a and b), the melt pool becomes unstable within the subsequent frames, and a liquid ligament is ejected from the explosion site (Supplementary Fig. 2c and d), leading to spatter formation (Supplementary Fig. 2e and f).

Supplementary Figure 2. Dynamics of spattering under the influence of subsurface explosion during electron beam scanning with an electron beam power of 426 W and a scan speed of 1.0 m/s. (a and b) X-ray images showing bubble explosions on front vapor depression wall during electron beam scanning. (c-f) X-ray images showing the dynamics of liquid ligament and spattering induced by bubble explosions during electron beam scanning. Prior to melting, a

preheating step was performed using the electron beam operated in pulse mode with a 0.4 duty cycle, a power of 318 W, a scan speed of 16 m/s, and a duration of 10 s. The duty cycle of a pulsed electron beam is the fraction of each pulse period during which the beam is actively on, defined as: $Duty\ cycle = \frac{t_{on}}{t_{on} + t_{off}}$, where t_{on} and t_{off} are the durations of the beam-on and beam-off periods within one pulse, respectively.

Fig. 6: The particle sizes visible in Fig. 6 do not match the powder particle size distribution of 15 to 45 μm stated in your methods section. In your radiographs, the particles seem to be roughly an order of magnitude larger than the used Al6061 powder. This indicates that the powder particles were actually molten during “sintering” which caused them to coalesce into larger particles. Please provide a detailed explanation on how the samples were prepared, including how the pre-sintered powder layers were created and applied to the thin-walled bulk samples used in your study.

Response 3-14: We thank the reviewer for the suggestion. Additional details about sample and sintered-layer preparation have been included in the revised manuscript, as also shown below in the end of this response.

The pre-sintered powder-bed Al6061 samples were prepared by sintering one layer of Al6061 powders (15–45 μm) on a commercial Al6061 plate using laser heating under an argon atmosphere to minimize oxidation. As observed in the X-ray images, the powder particles appear to have partially melted and coalesced into larger agglomerates, with apparent diameters of approximately 200 μm . This coalescence results from localized laser melting during the sintering process rather than the use of larger initial powder particles.

To avoid confusion for readers, clarification on sample and sintered-layer preparation and powder size was added to the revised manuscript, as shown below:

Lines 351-364: Al6061 was selected as the material for this study because of its good X-ray transparency for in-situ high-speed X-ray imaging experiments⁵². Two types of samples were prepared for the experiments: bare substrates and pre-sintered powder-bed samples. The bare substrate samples were sectioned from commercial Al6061 plates (purchased from McMaster). Since powder bed sintering is not the focus of this study, to accelerate the beamline experiments, the powder layer was pre-sintering before loading into the chamber. The pre-sintered powder-bed Al6061 samples were prepared by sintering one layer of Al6061 powders (15–45 μm) on a

commercial Al6061 plate using laser heating under an argon atmosphere to minimize oxidation. As observed in the X-ray images, the powder particles appear to have partially melted and coalesced into larger agglomerates, with apparent diameters of approximately 200 μm . This coalescence results from localized laser melting during the sintering process rather than the use of larger initial powder particles. Both types of Al6061 samples were cut into pieces with dimensions of 58 mm (length) x 10 mm (width) x 1.2 mm (thickness in the X-ray penetration direction), with the powdered surface facing the incident beam during experiments.

Discussion

L192-194: The explanation of bubble formation is highly flawed from a physical perspective. First, you essentially attribute the formation of a bubble to the fact that the irradiated material liquefies and evaporates below the surface first which results in an accumulation of vapor pressure that leads to the formation of a bubble. While it might be true that the energy deposition is highest a few μm below the irradiated surface, the outgoing heat flux (as a result of thermal diffusion) is also much higher there as the heated volume is surrounded by highly conductive material from all sides. In contrast, the surface of the irradiated material can only conduct heat downwards. This is why liquefaction of material irradiated by an electron beam should still be expected to start from its surface, unless specific energy transport calculations suggest otherwise. Second, your assumption that the material reaches its boiling point first where energy deposition rate is highest is incorrect. The boiling point of a material strongly depends on the surrounding pressure. This means that even if the material gets heated from the inside, it does not necessarily start to boil there as well (or rather evaporate) as it is still surrounded by solid or liquid material. Since the surface of the irradiated material is in contact with the vacuum environment of your process chamber, it requires significantly lower temperatures to evaporate (or rather has a much lower boiling point) and thus should be expected to do so first. This means that while your explanation for the observed bubble formation behavior is not necessarily incorrect, it is not convincing unless necessary calculations or simulations are performed that show that evaporation is highly likely to initiate inside the irradiated material which would likely be a totally unprecedented discovery.

Response 3-14: We thank the reviewer for this insightful and critical comment.

Regarding the first point on where melting initiates, we refer to the work by Wentao Yan *et al.* (*Computational Mechanics*, DOI 10.1007/s00466-015-1170-1), who modeled the temperature field in electron beam melting based on electron–atom interactions⁶. In their study, electron trajectories were tracked, and energy deposition was computed from electron–atom collisions for a Ti alloy (TC4) under a 60 kV acceleration voltage. The results showed that the maximum energy density occurs approximately 4.4 μm beneath the surface. During the initial formation of the melt pool, the subsurface region ($\sim 4.4 \mu\text{m}$ below the surface) attains the highest temperature first, reaching on the order of 3900–6000 K. (see Fig. 10 in Yan’s paper). Following the same physical mechanism, a similar subsurface temperature maximum is expected to form during the melting of Al alloys. However, it should be noted that the simulation in Yan’s study did not consider the presence of a gas phase; thus, bubble nucleation and explosion phenomena were not captured in their model. Future studies focusing on bubble formation, explosion, and their underlying causes are warranted.

We have revised our interpretation in the revised manuscript as follows:

Lines 204-207: Thus, as heating progresses, the temperature at the maximum energy deposition depth may reach the boiling point first. Simulation results by Yan *et al.* support this hypothesis, showing that under 60 kV electron-beam irradiation of Ti-6Al-4V, the subsurface region ($\sim 4.4 \mu\text{m}$ below the surface) reaches the highest temperature first, on the order of 3900–6000 K⁴³.

43. Yan, W., Smith, J., Ge, W., Lin, F. & Kam, W. Multiscale modeling of electron beam and substrate interaction: a new heat source model. *Comput. Mech.* 56, 265–276 (2015).

Regarding the second point on pressure and boiling temperature differences between the top surface and peak absorption layer, we performed a simple estimation. Let the pressure on the top melt pool surface be P_{top} , and the pressure at the peak absorption layer be P_{peak} . Then,

$$P_{\text{top}} = P_{\text{peak}} + \rho gh,$$

where ρ is the density of liquid aluminum ($\rho \approx 2.4 \times 10^3 \text{ kg/m}^3$), g is gravitational acceleration, and $h \approx 10 \mu\text{m}$ is the distance between the top surface and the peak absorption layer⁷. The resulting pressure difference, $\rho gh \approx 0.24 \text{ Pa}$.

During the electron beam melting experiments, as shown in the figures in our manuscript, a vapor depression is always present, indicating that evaporation does not occur in a true vacuum within the irradiated region. However, the vapor pressure during electron beam melting has not been reported in literature and is difficult to determine accurately.

Quantitative modeling and in situ diagnostics are necessary to further elucidate the mechanisms of subsurface bubble nucleation and explosion. Accordingly, we have clarified this point in the revised manuscript:

Lines 207-209: This localized overheating could initiate vaporization and the nucleation of subsurface bubbles filled with metal vapor (Fig. 3d). Future studies will focus on investigating the mechanisms underlying bubble formation.

Conclusion (Lines 343-347): This work is expected to inspire the electron beam additive manufacturing and welding community to advance modeling efforts for a quantitative understanding of bubble formation, explosion, and their effects, and to experimentally investigate, through in-situ monitoring, the occurrence of subsurface bubble explosions in high-atomic-number materials, thereby deepening the overall understanding of electron beam melting.

L.207-209 (+ L.214-217): You seem to essentially attribute all observed melt pool dynamics to the formation and explosion of bubbles with no consideration for other sources of hydrodynamic effects. Aside from the higher energy absorption efficiency and penetration depth provided by the electron beam, your experimental setup very closely resembles that of research groups from APS (Zhao et al. 2017, <https://doi.org/10.1038/s41598-017-03761-2>), SSRL (Calta et al. 2018, <https://doi.org/10.1063/1.5017236>) and DLS (Leung et al. 2018, <https://doi.org/10.1016/j.addma.2018.08.025>). These and some other groups have published extensive studies of melt pool dynamics in PBF-LB and should not be disregarded entirely on the basis of a difference in used heat source alone. Additionally, studies about melt pool dynamics in EBM from other research groups also exist and should be considered for discussion as well. The fact that your submitted article is entirely devoid of a comparison between your findings and the existing body of work in this field also seems questionable in general. I therefore highly recommend a discussion of your work that considers findings by other research groups, especially considering the similarities in observed melt pool dynamics (aside from the presented bubble formation and explosion mechanism). This should also result in a much more nuanced

understanding of the inducing effects of bubble formation and explosion discovered by your experiments.

Response 3-15: We sincerely thank the reviewer for this valuable and constructive comment. We agree that the observed melt pool dynamics cannot be solely attributed to bubble formation and explosion. Other hydrodynamic effects—including Marangoni convection, recoil pressure, surface tension gradients, and thermocapillary-driven flow—also play essential roles in shaping melt pool behavior. The references suggested by the reviewer are highly relevant and have been carefully reviewed and cited in the revised manuscript to strengthen the discussion.

In the updated *Results and Discussion* sections, we provide a more comprehensive analysis that situates our findings within the context of previous studies on melt pool dynamics under both laser and electron beam conditions. In addition to bubble explosion, we considered others factors that can influence the melt pool dynamics, including vapor pressure, evaporation, Rayleigh–Taylor instability and related mechanisms.

We have added the following discussion to the revised manuscript:

Regarding the evolution of the melt pool and keyhole:

Lines 112-116: At this stage, the rapid expansion of the melt pool and keyhole is primarily driven by rising vapor pressure within the depression, which displaces molten material and enhances heat flux into the surrounding solid, similar to the mechanism of keyhole drilling observed in laser powder bed fusion⁴⁶. Bubble explosions may further amplify local pressure fluctuations, promoting additional keyhole deepening.

46. Cunningham, R. et al. Keyhole threshold and morphology in laser melting revealed by ultrahigh-speed x-ray imaging. *Science* 363, 849–852 (2019).

Lines 232-233: These dynamics contribute to reshaping the melt pool and vapor depression through two coupled mechanisms.

Lines 256-261: During melting, recoil pressure from the dense metallic vapor drives the surrounding liquid outward, forming a deep and narrow cavity (Fig. 2i and j). As melting continues, the interplay among thermocapillary force, Marangoni convection, recoil pressure, and surface tension, similar to the mechanisms observed in laser powder bed fusion^{55,56}, induces keyhole

instability. When the keyhole collapses, the high-energy electron beam reinitiates intense evaporation in the newly formed shallow keyhole, driving a cyclic collapse–regrowth instability (Fig. 2k–o).

55. Qu, M. et al. Mitigating keyhole pore formation by nanoparticles during laser powder bed fusion additive manufacturing. *Addit. Manuf. Lett.* 3, 100068 (2022).

56. Zhao, C. et al. Critical instability at moving keyhole tip generates porosity in laser melting. *Science* 370, 1080–1086 (2020).

Regarding spattering:

Lines 165-177: The dynamics of spattering under the influence of subsurface bubble formation and explosion were also investigated. Since spatter primarily arises from interactions between metallic vapor and the melt pool surface or powder bed, pre-sintered powder bed samples were used to study interactions among metal vapor, melt pool and powder. Following subsurface bubble explosions (Fig. 6a and b), the melt pool became unstable within the subsequent frames, ejecting a liquid ligament from the explosion site followed by spatters (Fig. 6c-f and Supplementary Movie 7). To facilitate clearer identification of bubble explosions and the associated spatters, the original images were processed and are shown in Fig. 6g and h. An additional example of bubble-explosion–induced spattering with preheating prior to melting is provided in Supplementary Note 2, Supplementary Fig. 2 and Supplementary Movie 8. It is noted that strong evaporation at the vapor depression without bubble explosion can also drive spatter ejection, as supported by numerous X-ray studies in both laser and electron beam powder bed fusion processes^{34,47}. The bubble explosions observed here represent an additional mechanism that can contribute to spatter formation.

34. Semjatov, N., Wahlmann, B. & Carolin, K. Multiple interaction electron beam powder bed fusion for controlling melt pool dynamics and improving surface quality. *Addit. Manuf.* 90, 104316 (2024).

47. Guo, Q. et al. Transient dynamics of powder spattering in laser powder bed fusion additive manufacturing process revealed by in-situ high-speed high-energy x-ray imaging. *Acta Mater.* 151, 169–180 (2018).

Regarding melt flow:

Line 300-305: Second, the backward melt flow observed in Fig. 9c, a phenomenon commonly reported in laser powder bed fusion, typically arises as molten metal circulates around the vapor depression and gains momentum opposite to the scanning direction. Bubble explosions occurring at the front wall may introduce additional momentum to the backward flow by generating transient vapor jets and surface waves that propagate along the vapor–liquid interface.

Lines 306-313: As the melt flow propagates backward, periodic undulations develop along the melt pool surface due to the Rayleigh–Taylor instability, as previously reported in laser and electron beam powder bed fusion^{34,41}. As the solidification front advances, these undulations evolve into humping structures along the melt track. In electron beam melting, intermittent momentum impulses generated by bubble explosions further amplify this instability by perturbing the melt pool surface. The combined effects of Rayleigh–Taylor and explosion-induced instabilities thus promote periodic undulations, ultimately leading to the formation of humping structures along the melt track (Fig. 9d).

34. Semjatov, N., Wahlmann, B. & Carolin, K. Multiple interaction electron beam powder bed fusion for controlling melt pool dynamics and improving surface quality. *Addit. Manuf.* **90**, 104316 (2024).

41. Bhatt, A. et al. In situ characterisation of surface roughness and its amplification during multilayer single-track laser powder bed fusion additive manufacturing. *Addit. Manuf.* **77**, 103809 (2023).

L.237- 239: Is the vapor plume inside the depression actually strong enough to sufficiently scatter the electron beam to induce collapse? Be reminded that the high acceleration voltage is sufficient to allow the electron beam to penetrate fairly deep into solid material. How about performing a rough calculation of the penetration depth of the electron beam for reasonable vapor pressures, based on the same equations you used to calculate the energy deposition profile in Fig. 7a (by assuming that the vapor has the same composition as the solid, just at a much lower density)? This would contextualize your conclusion by some actual numbers (even if they are based on estimates). Additionally, your explanation that electron scattering causes keyhole collapse does not match the general observation that keyholes start to collapse from the surface through the

formation of a constriction. How about considering material transport phenomena inside the melt as well?

Response 3-16: We appreciate the reviewer’s insightful suggestions.

To assess the plausibility of electron beam scattering by the vapor plume, we estimated the beam penetration depth under vapor-phase conditions.

First, we used the ideal gas law $\rho = \frac{PM}{RT}$ to estimate the density of gaseous aluminum. The molar mass of aluminum is $M = 0.02698 \frac{kg}{mol}$, the gas constant $R = 8.314 J/(mol \cdot K)$, and the temperature is taken as aluminum’s boiling point, $T = 3013 K$.

As we mentioned, the vapor pressure during electron beam melting has not been reported in literature and is difficult to determine accurately. Here, we use reported vapor pressures from laser powder bed fusion as a rough order of magnitude estimate, while acknowledging that future work is needed to quantify the differences between laser and electron beam melting conditions. In laser powder bed fusion, the recoil pressure P_{recoil} is on the order of 50–90 kPa (Cullom T. *et al.* Frequency domain measurements of melt pool recoil force using modal analysis. *Sci. Rep.* **11**, 1–11 (2021)). Under EBM’s low-pressure (vacuum) environment, the vapor pressure is expected to be lower than 50 kPa. Using this value yields an approximate vapor density:

$$\rho_{vapor}(50 \text{ kPa}) < 0.054 \text{ kg/m}^3$$

Compared with the solid aluminum density ($\rho_{vapor} \approx 2700 \text{ kg/m}^3$), the density ratio exceeds 5.0×10^4 . This indicates that solid aluminum is more than four to five orders of magnitude denser than its vapor phase.

The penetration depth D_{solid} of 60 keV electrons in solid aluminum is approximately 30 μm (Klassen, A., Bauereiß, A. & Körner, C. Modelling of electron beam absorption. *J. Phys. D Appl. Phys.* **065307**, 065307 (2014)). Assuming penetration depth scales inversely with material density, the corresponding penetration depth in aluminum vapor can be estimated as:

$$D_{vapor} \approx D_{solid} \times \frac{\rho_{solid}}{\rho_{vapor}} > 30 \times 10^{-6} \text{ m} \times 5.0 \times 10^4 \approx 1.5 \text{ m.}$$

This estimated range greatly exceeds the typical keyhole depth (~200 μm), suggesting that electron scattering by vapor alone is insufficient to induce keyhole collapse. Therefore, the observed collapse is more plausibly attributed to fluid-dynamic instabilities within the melt—such as recoil pressure fluctuations, Marangoni convection, and surface-tension-driven flow—rather than direct beam–plume interactions. Future quantitative analysis using Monte Carlo electron transport simulations could further refine these estimates.

In response, we have carefully revised the related discussion in the *Results* section (Lines 274–279) to present a more physically grounded interpretation of the keyhole collapse mechanism. The revised text now reads:

Lines 256-261: During melting, recoil pressure from the dense metallic vapor drives the surrounding liquid outward, forming a deep and narrow cavity (Fig. 2i and j). As melting continues, the interplay among thermocapillary force, Marangoni convection, recoil pressure, and surface tension, similar to the mechanisms observed in laser powder bed fusion, induces keyhole instability (Fig. 2k). When the keyhole collapses, the high-energy electron beam reinitiates intense evaporation in the newly formed shallow keyhole, driving a cyclic collapse–regrowth instability (Fig. 2k–o).

L.285-287: The backwards melt flow visible in Fig. 9 is a fairly common phenomenon that does not rely on bubble explosion to occur. The backwards material ejection simply follows from the molten material having to flow around the depression, during which it gains momentum in the opposite direction of the moving electron beam/depression. Like previously stated, don't just attribute all observations to bubble formation and explosion without considering more commonly known and often likely alternatives.

Response 3-17: We thank the reviewer for this insightful comment. We agree that backward melt flow is a commonly observed phenomenon in laser powder bed fusion processes, and it can indeed arise from hydrodynamic flow around the vapor depression as the molten material gains momentum opposite to the beam scanning direction. In our revised manuscript, we have clarified that bubble explosion can introduce additional momentum to the backward flow.

Lines 300-305: Second, the backward melt flow observed in Fig. 9c, a phenomenon commonly reported in laser powder bed fusion, typically arises as molten metal circulates around the vapor

depression and gains momentum opposite to the scanning direction⁵⁷. Bubble explosions occurring at the front wall may introduce additional momentum to the backward flow by generating transient vapor jets and surface waves that propagate along the vapor–liquid interface.

57. Guo, Q. et al. In-situ full-field mapping of melt flow dynamics in laser metal additive manufacturing. *Addit. Manuf.* 31, 100939 (2020).

L.293: Humping is a well known phenomenon in PBF-LB, where based on your provided explanation, bubble formation is not possible to occur. For reference, see Bhatt et al. 2023 (<https://doi.org/10.1016/j.addma.2023.103809>) for example.

Response 3-18: We thank the reviewer for this insightful comment and for pointing out the relevant reference. We agree that humping in laser powder bed fusion, where bubble formation does not occur, primarily arises from Rayleigh–Taylor instability. Our revised text now acknowledges this established mechanism and clarifies that bubble explosions are not the sole cause of humping in electron beam melting (EBM). Instead, we propose that in EBM, bubble-induced transient pressure pulses act as an additional source of perturbation, superimposing on the inherent Rayleigh–Taylor instability. The revised paragraph (Lines 324–331) now reads as follows:

Lines 306-313: As the melt flow propagates backward, periodic undulations develop along the melt pool surface due to the Rayleigh–Taylor instability, as previously reported in laser and electron beam powder bed fusion^{34,41}. As the solidification front advances, these undulations evolve into humping structures along the melt track. In electron beam melting, intermittent momentum impulses generated by bubble explosions further amplify this instability by perturbing the melt pool surface. The combined effects of Rayleigh–Taylor and explosion-induced instabilities thus promote periodic undulations, ultimately leading to the formation of humping structures along the melt track (Fig. 9d and Supplementary Movie 5).

34. Semjatov, N., Wahlmann, B. & Carolin, K. Multiple interaction electron beam powder bed fusion for controlling melt pool dynamics and improving surface quality. *Addit. Manuf.* **90**, 104316 (2024).

41. Bhatt, A. et al. In situ characterisation of surface roughness and its amplification during multilayer single-track laser powder bed fusion additive manufacturing. *Addit. Manuf.* 77, 103809 (2023).

L.308: Your statement that the bubble formation mechanism is not material-specific is incorrect, especially considering that there is no evidence for it. While electron beams can penetrate all metallic materials, the extent of their ability to do so differs drastically between alloys. Al6061 is arguably the material exhibiting the most amount of electron beam penetration (of all commonly processed EBM alloys) due to its low density (and low average atomic number). More commonly processed materials in EBM, like Cu, steels, Ni-base superalloys and refractory alloys offer much lower beam penetration depths and have (if your provided explanation for bubble formation is assumed to be correct) therefore a much lower chance of exhibiting bubble formation. Even a previously published article by the same group (Yuan et al. 2024, <https://doi.org/10.1016/j.addlet.2024.100239>) where Ti6Al4V and Al6061 was used to perform in-situ X-ray imaging experiments, shows no signs of bubble formation and explosion. This suggests that the observed phenomenon is, contrary to stated, highly material specific and likely unique to the combination of Al6061 and EBM and that it does not occur all the time, as Al6061 was melted in this study using the same sample geometry, beam power and scan velocity without exhibiting the bubble formation phenomenon (other melt pool dynamics seem to be similar to those reported here though).

Response 3-19: We thank the reviewer for this insightful and detailed comment. We agree that the bubble formation and explosion behavior should be material-specific due to differences in physical properties, such as melting point, boiling point, atomic number, and density, which can strongly influence electron beam energy absorption, penetration depth, and vaporization behavior during EB-PBF. The high beam penetration depth and low boiling temperature of Al6061 make it more susceptible to subsurface vaporization and bubble formation compared with denser materials with high boiling points, such as Cu, Ni-based superalloys, or steels.

Further experimental studies on other alloy systems (e.g., Cu, Ni-based superalloys) are necessary to determine the generality of the observed bubble explosion phenomenon and its dependence on material properties. To reflect this important point, we have revised the manuscript *Title, Abstract, Introduction, Discussion, and Conclusion* to focus specifically on aluminum alloys and to clarify that the present findings are material-dependent, while acknowledging the need for further validation across different material systems.

Title (Lines 1 and 2): Bubble explosion induced melt pool instabilities in electron beam melting of aluminum alloy

Abstract (Lines 19-21): Here, using in-situ high-speed synchrotron X-ray imaging, we reveal that bubble explosions, unique to EBM, induce melt pool instabilities contributing to defect formation in Al6061.

Introduction (Lines 58-60): In this study, taking advantage of this in-situ characterization system, we discovered that bubble explosions during EBM in Al6061, unique to EBM, destabilize the melt pool and contribute to multiple types of defects.

Discussion: We deleted the previous statement: *“The melt pool instabilities induced by bubble explosion originate from the energy deposition characteristics inherent to the high-energy electron beam. This mechanism is not material-specific and is expected to manifest across all material systems. However, the energy absorption behavior, including the electron beam peak energy deposition depth and penetration depth, is highly material-dependent.”*

Conclusion (Lines 343-347): This work is expected to inspire the electron beam additive manufacturing and welding community to advance modeling efforts for a quantitative understanding of bubble formation, explosion, and their effects, and to experimentally investigate, through in-situ monitoring, the occurrence of subsurface bubble explosions in high-atomic-number materials, thereby deepening the overall understanding of electron beam melting.

We appreciate the reviewer’s careful observation. In that earlier work, our primary focus was on the formation of defects that directly evolve into porosity during electron beam melting. At that stage, our intent was not to present or interpret all possible phenomena occurring in electron beam melting. Therefore, we limited the scope of the discussion and did not include the full sequence of subsurface bubble formation and explosion in that paper.

In the present study, we systematically investigate and report this previously unaddressed phenomenon of subsurface bubble formation and explosion in detail. The difference in focus, rather than in experimental outcome, explains the absence of explicit bubble evolution images in our earlier publication.

Moreover, even in our earlier work, there are subtle indications of bubble-related events. For example, in Fig. 3b and c of Yuan et al. (2024), ripples can be observed along the rear wall of the keyhole. These features are explosion-induced surface waves, showing a similarity to the ripples observed in Figs. 4 and 5 of the present manuscript. These consistencies suggest that while the previous study did not explicitly discuss bubble dynamics, the underlying phenomena were indeed present.

L.314-316: The high beam penetration depth of EBM is arguably its biggest advantage over other additive manufacturing techniques, as it leads to a much better distribution of heat and lower evaporation during melting. Suggesting to remove this benefit to avoid bubble formation is highly counter-intuitive and would defy the purpose of EBM almost entirely.

Response 3-20: We thank the reviewer for this insightful comment. We agree that the high beam penetration depth is indeed a key advantage of EBM, as it enables a more uniform heat distribution and reduces excessive evaporation during melting. We have removed the discussion suggesting accelerating voltage adjustment (Page 16, Line 314 in the original manuscript) in the revised manuscript.

Methods

L.333: Was any X-ray transparent confinement used, between the Al6061 samples were sandwiched? If not, did the melt pool stay confined between solid Al6061 during melting (i.e. did you make sure to create a melt pool thinner than the 1.2 mm sample thickness). Could you include an image of an Al6061 sample post experimental procedure in Fig. 1, such that the reader gets an expression of how melting affects the sample?

Response 3-21: No confinement was used during the melting experiments. The thickness of the Al6061 samples was 1.2 mm. During melting, the melt pool remained confined within the boundaries of the Al6061 sample, ensuring the formation of a fully developed melt pool without penetrating the 1.2 mm sample thickness. An optical image of the melt track has been added to Fig. 1, and the setup for the optical tests has been included in the *Materials and Methods* section.

Lines 68-71: During melting, the melt pool remained confined within the boundaries of the Al6061 sample, ensuring the formation of a fully developed melt pool without penetrating the 1.2 mm sample thickness (a representative surface profile of the melt track is shown in Fig. 1b).

Fig. 1. In-situ electron beam melting high-speed synchrotron X-ray imaging experiment. (a) Illustration of the experimental setup of in-situ high-speed synchrotron X-ray imaging of electron beam melting. **(b)** Typical surface profile of the melt track after electron beam melting. High-speed X-ray imaging with a frame rate of 50 kHz, 100 kHz and 135 kHz, and spatial resolution of 2 μm was used to capture the dynamics of vapor depression, melt pool and spatter during electron beam melting. The experiments were conducted in a vacuum chamber using a beam with a diameter of approximately 200 μm (full width at half maximum (FWHM) of a Gaussian energy distribution).

L.342-343: As previously stated, could you please provide a spot size definition for your electron beam diameter of 200 μm ?

Response 3-22: We thank the reviewer for this insightful comment. In the experiments, the electron beam has a diameter of approximately 200 μm at the working plane, defined as the full width at half maximum (FWHM) of a Gaussian energy distribution. The beam information was measured using a slit-scan method, in which the beam was stepped across a slit of known width while measuring the transmitted current to determine the rise/fall regions. Then the spot size was

calculated using the FWHM metric. This clarification has been incorporated into the revised manuscript to ensure accuracy and consistency in the beam specification.

Lines 80-82: The experiments were conducted in a vacuum chamber using a beam with a diameter of approximately 200 μm (full width at half maximum (FWHM) of a Gaussian energy distribution).

Lines 371-376: In the experiments, the electron beam has a diameter of approximately 200 μm at the working plane, defined as the full width at half maximum (FWHM) of a Gaussian energy distribution. The beam information was measured using a slit-scan method, in which the beam was stepped across a slit of known width while measuring the transmitted current to determine the rise/fall regions. Then the spot size was calculated using FWHM metric.

References

L.448-449: Reference 34 (Bidola, P. M. et al. In situ synchrotron...) is wrongly referenced. The first author of this publication should be “Semjatov, N.” not “Bidola, P. M.”.

Response 3-23: The mistake has been corrected.

Lines 487-489 Semjatov, N. et al. In-situ synchrotron imaging of powder consolidation and melt pool dynamics in electron beam powder bed fusion. Addit. Manuf. 110, 104943 (2025).

Final remark

Assuming the described methodology is complete, the in-situ synchrotron imaging study was performed at room temperature. In EBM, processes are always performed at an elevated build temperature to enable sintering of powder particles with minimal energy input, in order to minimize the risk of smoke and reduce accumulation of thermal stress. As a result, solidification times are usually a few orders of magnitude larger than in PBF-LB which is one of the reasons why melt pool dynamics can differ drastically between the two techniques. Since elevated temperatures were not used in your study, solidification behavior should actually be more similar to that encountered in PBF-LB than in EBM (PBF-EB). How do you think does this affect the

transferability of your findings to industrial PBF-EB processes? Would you expect to see the same melt pool dynamics, bubble formation and explosion behavior, as well as humping formation or would you expect differences?

Response 3-24: We agree with the reviewer that electron beam melting is always performed with preheating. Electron beam melting with preheating was also conducted during our beamline experiments. Preheating was carried out by electron beam scanning with a lower electron beam power and a fast speed before melting. Subsurface bubble formation and explosions were observed, accompanied by similar melt pool dynamics (Supplementary Note 3 and Supplementary Fig. 3). The figures and corresponding video of melting Al6061 with preheating have been added to the supplementary materials.

Our findings are relevant to industrial PBF-EB processes. The fundamental mechanisms revealed in this study—such as subsurface bubble formation, vapor depression dynamics, and melt pool instability—are governed by the same physical principles that operate in large-scale industrial systems. Although the beam parameters, scanning conditions and sample dimensions may differ, the underlying electron–matter interactions are similar. Therefore, the insights gained here are relevant for understanding and optimizing industrial PBF-EB processing conditions.

The following contents have been added to the revised manuscript and supplementary materials:

Lines 326-329: The influence of preheating on melting dynamics, including subsurface bubble formation, melt pool behavior, and vapor depression evolution was also investigated. Subsurface bubble formation and explosions were observed, accompanied by similar melt pool dynamics (Supplementary Note 3 and Supplementary Fig. 3).

Supplementary note 3: Influence of preheating on melting dynamics

The influence of preheating on melting dynamics, including subsurface bubble formation, melt pool behavior, and vapor depression evolution was also investigated. Preheating was carried out by electron beam scanning with a lower electron beam power and a fast speed before melting. Subsurface bubble formation and explosions were observed, accompanied by melt pool dynamics similar to those observed without preheating (Supplementary Fig. 3a). Driven by the Plateau-Rayleigh and explosion-induced instabilities, this elongated liquid column (i.e., the melt pool)

eventually forms a humping structure on the melt track after solidification (Supplementary Fig. 3b-d).

Supplementary Fig. 3. Melting dynamics during electron beam scanning with preheating. (a-c) X-ray images showing the bubble explosion and melt pool dynamics during electron beam scanning. **(d)** X-ray image showing periodic humping after solidification. Prior to melting, a preheating step was performed using the electron beam operated in pulse mode with a 0.4 duty cycle, a power of 318 W, a scan speed of 16 m/s, and a duration of 10 s. Melting was then performed with an electron beam power of 426 W and a scan speed of 1.0 m/s.

Aside from my remarks, I find the presented experimental findings highly interesting and am looking forward to their publication once the necessary adjustments have been made.

Sincerely,

Nick Semjatov

Response to Reviewer's comments

We deeply appreciate the editor's and reviewers' time and effort in providing constructive comments and valuable advice on our manuscript. We have made our best efforts to address all the insightful comments. Below, we provide a detailed description of the corrective actions and our point-by-point responses to each reviewer's comments.

REVIEWERS' COMMENTS

Reviewer #1 (Remarks to the Author):

The authors responded well and made revisions to the reviewer's comments. The newly revised paper focuses on Al6061 aluminum alloys, which is more rigorous and comprehensive, and also provides reference and inspiration for EBM processes of other materials. Therefore, I believe that the paper can be accepted.

Response 1-1: We sincerely thank the reviewer for the thorough evaluation of our manuscript and the encouraging recommendation for acceptance.

Reviewer #2 (Remarks to the Author):

The authors have addressed the reviewers' comments in an appropriate and satisfactory way. In particular, the hedging of the interpretations of the observations to the actual experiment improve the manuscript. Further the clarification of the phenomenological nature, as well as the identification of the limitations and scope of the work benefit the manuscript.

The presented results and their interpretation are more clear and categorized. The limitations of the work and the directions of future work are pointed out, which strengthens the contribution of the work to PBF-EB research.

Response 2-1: We thank the reviewer for the positive and constructive feedback. We appreciate the reviewer's recognition that the revisions have improved the clarity, scope, and overall contribution of the manuscript.

The authors should include some markers/guidance to the eye to ensure that there is no confusion about the observations in the videos.

With including this adjustment, the manuscript gives a clear and coherent presentation of phenomenological observations and situates these in the research field. It is therefore suitable for consideration for publication.

Response 2-2: We thank the reviewer for this helpful suggestion. We have added visual markers and guidance cues to the videos to highlight key observations, such as the melt pool boundary, vapor depression, and bubble dynamics.

Reviewer #3 (Remarks to the Author):

This revision leaves the manuscript in a much better state. All of my remarks have been sufficiently addressed. With the inclusion of video material, the authors provide much stronger and easier to follow evidence for their reported observations (the bubble formation and explosion phenomenon is much easier to see this way). The description of experimental procedures is also significantly improved and the description of scientific impact and relevance to the field of PBF-EB appropriately adjusted. There is no further revision needed from my point of view. I am looking forward to the publication of this study.

Sincerely,

Nick Semjatov

Response 3-1: We sincerely thank the reviewer for the very positive and encouraging assessment. We greatly appreciate the reviewer's recognition of the improved presentation and relevance of the work to the field of PBF-EB research.